# Evaluating Sample Utility for Efficient Data Selection by Mimicking Model Weights

**Tzu-Heng Huang** [1 2]  **Manjot Bilkhu** [2]  **John Cooper** [1]  **Frederic Sala** [1]  **Javier Movellan** [2]

## Abstract

Large-scale web-crawled datasets contain noise, bias, and irrelevant information, necessitating data selection techniques. Existing methods depend on hand-crafted heuristics, downstream datasets, or require expensive influence-based computations—all of which limit scalability and introduce unwanted data dependencies. To address this, we introduce the Mimic Score, a simple and geometry-based data-quality metric that evaluates utility by measuring alignment between a sample's gradients and a target direction induced by a pre-trained reference model. This leverages readily available model weights, avoids needing validation datasets, and incurs minimal computational overheads. Building on this metric, we propose Grad-Mimic, a two-stage framework that re-weights samples online to accelerate training and aggregates sample utilities offline to construct effective data filters. Empirically, we show that using mimic scores to guide training improves data efficiency, accelerates convergence, yields consistent performance gains across six image datasets, and enhances CLIP models with 20.7% fewer training steps. Additionally, mimic score-based filters augment existing filtering techniques, enabling improved CLIP models trained with 4.7 million fewer samples. Our code is released here.

## 1. Introduction

Large-scale web-crawled datasets are fundamental to the success of modern multimodal models such as AIMv2 (Fini et al., 2024), SigLIP2 (Tschannen et al., 2025), OpenAI CLIP (Radford et al., 2021), and ALIGN (Jia et al., 2021). These datasets provide vast quantities of information—but

inevitably contain noise, biases, or misleading content. To mitigate these, data selection—ruling out undesirable samples—has emerged as a critical step in the development pipeline (Albalak et al., 2024; Bai et al., 2024). For example, the FineWeb dataset, containing 15 trillion text tokens, undergoes eight carefully designed filtering steps to enhance model performance (Penedo et al., 2024).

Current selection strategies; however, are far from satisfactory. We can broadly divide them into two categories: model-free methods, such as those based on hand-crafted heuristics or semantic similarity to downstream datasets (Xie et al., 2023; Wang et al., 2024b) often *require costly trial-and-error* to establish selection rules and *add unwanted data dependencies*. These methods also lack the granularity needed to assess individual utility, leading to suboptimal training outcomes. In contrast, model-based techniques, such as those building specialized filtering networks (Fang et al., 2023; Thakkar et al., 2023; Chen & Mueller, 2024; Wettig et al., 2024; Wang et al., 2025), scoring samples through a reference model loss (Mindermann et al., 2022; Lin et al., 2024b), or using influence functions (Yu et al., 2024; Wang et al., 2024a; Xia et al., 2024; Wu et al., 2024), *increase pipeline complexity* and are often *computationally expensive at scale.* These shortcomings motivate the need for a *simpler, dataset-agnostic, compute-efficient* mechanism for identifying high-value data.

We introduce the ***Mimic Score***, a simple data-quality metric that quantifies each training sample's contribution to effective weight updates. We argue that samples whose gradients potentially pull the model in undesirable directions—and thus misguide weight updates—should be down-weighted or discarded. To establish a desirable weight update, we leverage high-quality artifacts currently public: *pre-trained weights*, located in a more optimal part of the weight space, as *our proxy*. The mimic score is computed for each sample by measuring the alignment between its negative gradient and a direction pointing to this reference model. A high alignment score indicates a sample can steer the model toward this preferred region; a low one diverts it.

Using weight-space geometry to evaluate sample utility offers multiple benefits. **First**, the prevailing ***access asymmetry*** in the modern AI landscape makes this approach more

[1]University of Wisconsin-Madison [2]Apple Inc.. Correspondence to: Manjot Bilkhu <mbilkhu@apple.com>, Tzu-Heng Huang <thuang273@wisc.edu>.

*Proceedings of the 43rd International Conference on Machine Learning*, Seoul, South Korea. PMLR 306, 2026. Copyright 2026 by the author(s).

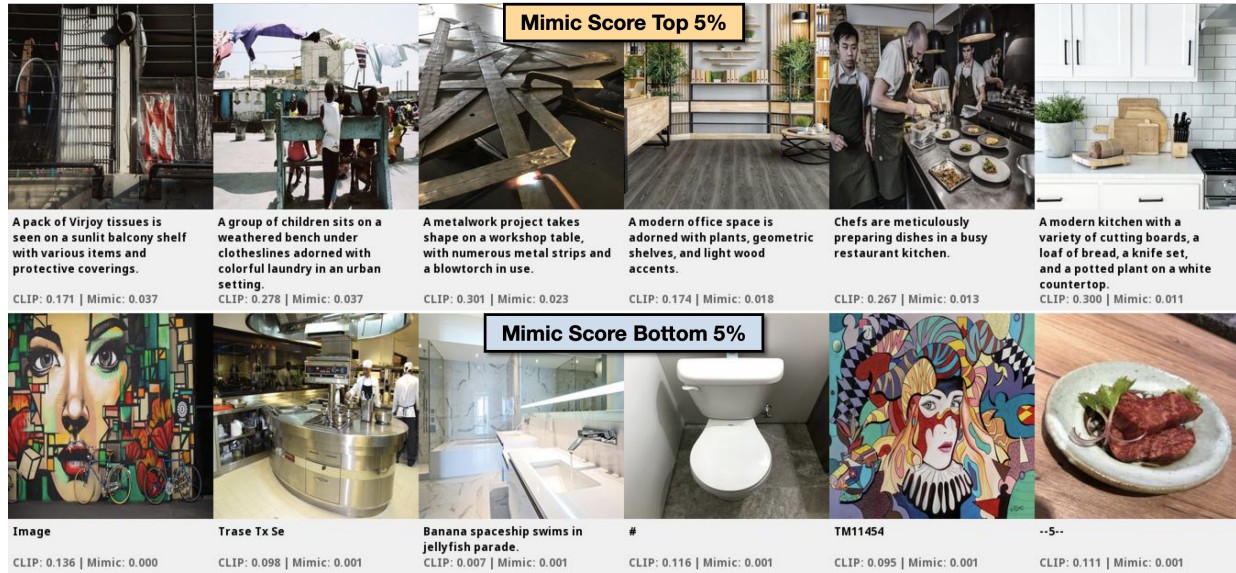

*Figure 1.* **High-value vs. low-value samples identified by Mimic Score**: Random samples from the top 5% (first row) and bottom 5% (second row) of web-crawled data, ranked by their mimic score. Each image includes its caption and CLIP score (higher indicates better quality). Mimic scores align closely with CLIP scores. High-value samples generally have detailed captions and coherent visuals, while low-value ones carry short captions and misaligned content.

practical than ever: general-purpose well-trained models are increasingly released to the public, but the carefully curated datasets used to produce them remain proprietary. **Second**, unlike methods that rely on gradients computed over a clean validation split (Wang et al., 2024a; Xia et al., 2024; Wu et al., 2024), mimic score *avoids extra validation computations, bypassing the difficulties of curating such datasets and ensuring its quality*. **Third**, many reference model-based techniques compute loss difference through additional model inference (Mindermann et al., 2022; Lin et al., 2024b). Our mechanism instead is a much simpler operation based on the discrepancy of model weights, which *substantially reduces compute overheads*.

Building on this metric, we develop ***Grad-Mimic***, a two-stage data selection framework:

- **Online Batch Re-weighting.** During training, Grad-Mimic computes mimic scores on the fly and prioritizes samples to learn, enhancing data efficiency and accelerating convergence.
- **Offline Sample Selection.** After training, Grad-Mimic aggregates per-sample's mimic scores across training steps to estimate an overall sample utility. Such combined estimate serves as a reliable filter, enabling the construction of a smaller, higher-quality dataset for future use.

We validate the capabilities of Grad-Mimic in a broad range of scenarios. For **mislabeled sample detection**, we show in a controlled setting, Grad-Mimic accurately identifies noisy samples, and the derived mimic scores correlate to overall dataset quality, *achieving a Pearson correlation of 0.903*. By down-weighting undesirable samples during training, Grad-Mimic consistently improves data efficiency and model performance across six image datasets, while reducing runtime overheads by $2.6\times$ compared to competing methods. For **large-scale data curation**, we scale our testbed to web-crawled data by training CLIP models from scratch at DataComp scale (Gadre et al., 2023) (e.g., 10M+ and 100M+ samples). Using pre-trained weights as our reference, mimic scores effectively navigate training, *reducing 20.7% training steps to convergence*. The resulting filter complements existing selection strategies, improving CLIP model performance with *4.7M fewer training samples*.

We summarize our contributions as follows:

- **A New Data-quality Metric**: We exploit access asymmetry and introduce the Mimic Score, which distills weight-space geometry into data-quality insights.
- **An Efficient Data Selection Framework**: We present Grad-Mimic, a framework that prioritizes high-value samples to learn, enhances data efficiency, accelerates convergence, and automates effective data selection.
- **Substantial Computational Gains.** Compared to prior model-based methods, the mimic score computation incurs lower runtime and reduced GPU memory usage.
- **Reliable Ensemble Filtering.** Aggregated mimic scores provide reliable estimates of dataset quality and model performance gains, and further strengthen existing filters.
- **Extensive Ablation Studies.** We provide extensive studies, studying our framework generalization under diverse design choices and the impact of reference model quality.

## 2. Related Work

Our work explores using weight-space geometry to evaluate sample utility, intersecting three key areas: **(i)** data selection, **(ii)** multimodal data curation, and **(iii)** weak supervision.

**Data Selection.** Data selection techniques can be categorized into group- and sample-level approaches. Group-level approaches focus on optimizing domain mixtures to curate high-quality datasets (Fan et al., 2023; Xie et al., 2024; Chen et al., 2024; Ge et al., 2025). Sample-level methods, which are the focus of Grad-Mimic, aim to provide a score measuring sample utility and rule out undesired samples. Prior research has explored various gradient-based strategies, such as identifying impactful samples through gradient magnitudes (Paul et al., 2021), analyzing gradient similarities across training batches (Sedova et al., 2023), selecting key samples that can capture full training (Killamsetty et al., 2021; Mirzasoleiman et al., 2020), and using sample influence computed from noise-free validation sets (Wang et al., 2024a; Xia et al., 2024; Wu et al., 2024). These methods; however, are sensitive to noisy batch gradients and often incur extra computation. *We sidestep both with the help from a reference model and measure sample's directional compatibility in the weight-space geometry.*

**Data Curation for Multimodal Models.** Multimodal models rely on large-scale web datasets (Thomee et al., 2016; Schuhmann et al., 2022; Gadre et al., 2023); however, quantity comes at a price: unconstrained crawling amplifies noise and requires careful curation. Previous approaches have included selecting samples based on semantic similarity to the downstream evaluation datasets (Xie et al., 2023; Wang et al., 2024b; Thrush et al., 2024), using semantic deduplication (Abbas et al., 2023), training specialized filtering networks (Fang et al., 2023; Chen & Mueller, 2024; Wettig et al., 2024; Thakkar et al., 2023), and using data influence functions (Lin et al., 2024a; Yu et al., 2024). While effective, they require access to other datasets, introduce training complexities, or demand substantial compute. Grad-Mimic offers a more efficient alternative: *distilling readily available weights into data insights for sample identification.*

**Weak Supervision.** Weak supervision is a framework to construct labeled datasets by combining multiple noisy label estimates (Ratner et al., 2016; 2017; 2019; Fu et al., 2020). These estimates can come from sources such as heuristic labeling rules, domain knowledge, or pre-trained models (Huang et al., 2024a; 2023a), often encoded as labeling functions. Labeling function outputs are modeled and aggregated to produce a probabilistic labeling decision. Weak supervision has demonstrated success in diverse domains (Hooper et al., 2020; Fries et al., 2019; Khattar et al., 2019; Roberts et al., 2022; Shin et al., 2021). Prior works focus on weak label aggregation to construct datasets. We adopt it but for a new purpose: *modeling accumulated sam-*

*ple utility assessments for a combined selection decision.*

## 3. Evaluating Sample Utility

Our goal is to quantify training sample contributions to learning process when we **(i)** have a new model to be trained and **(ii)** a well-trained reference model is available. Our principle is that *samples that potentially pull the model in an undesirable direction, thereby misdirecting weight updates, should be considered low-value.* We start with notation, then explain how an intermediate scoring metric is derived and used inside our framework, Grad-Mimic (Sec. 4).

**Notation.** Let $D = \{s_i\}_{i=1}^n$ denote a dataset of $n$ samples drawn from a distribution $\mathcal{D}$ supported on the space $\mathcal{S}$. At training step $t$, model parameters $\theta_t$, are iteratively optimized using the dataset $D$. While our framework supports various training settings, we focus on supervised learning for clarity. We assume $\mathcal{S} = \mathcal{X} \times \mathcal{Y}$, where $\mathcal{X}$ is the input space and $\mathcal{Y}$ is the label space. Each sample $s_i$ can be expressed as $(x_i, y_i)$, where noise may be present either in the instance $x_i$ or in the label $y_i$. The empirical loss across all the samples is defined as $\frac{1}{n}\sum_{i=1}^n \ell(x_i, y_i)$, and each sample's gradient with respect to the model parameters $\theta_t$ is written as: $g_{i,t} := \nabla_{\theta_t} \ell(x_i, y_i)$. A standard update to model parameters is $\theta_{t+1} := \theta_t - \eta \frac{1}{b}\sum_{i=1}^b g_{i,t}$, where $\eta$ is the learning rate and $b$ is the batch size.

**Mimic Score Calculation.** To evaluate whether a sample's gradient pulls the model in an undesirable direction, we use the geometry information from a pre-trained model's weights as our proxy guide. These reference weights, denoted by $\theta_{\text{ref}}$, are assumed to reside in a more optimal part of the weight space, such that $\ell(\theta_{\text{ref}}) < \ell(\theta_t)$. We use the vector from the current model's weight space $\theta_t$ to $\theta_{\text{ref}}$ to measure each sample utility in approximating a better weight configuration ($\theta_{\text{ref}}$). In practice, these weights can be layer-specific, e.g., model weights in the last layer, which usually store more informative features than earlier layers (Ghiasi et al., 2022; Gandelsman et al., 2023).

Reference weights can be obtained either by (i) training a model on a relatively small, hold-out dataset until it meets the desired performance (Lin et al., 2024b; Mindermann et al., 2022), if resource permits, or working on new domains where no pre-trained weights exist; (ii) or, more efficiently, by using an existing well-trained public model, which can sidestep *both data-access constraints and costly training.*

At each training step $t$, we define a *time-varying* vector that points toward the reference model weights, given by $v_t := \theta_{\text{ref}} - \theta_t$. We examine how each sample's negative gradient $-g_{i,t}$, intended for updating model weights, aligns with vector $v_t$. We measure this alignment degree by considering both the direction and strength of the negative gradient. Specifically, we compute the projection length of $-g_{i,t}$ onto

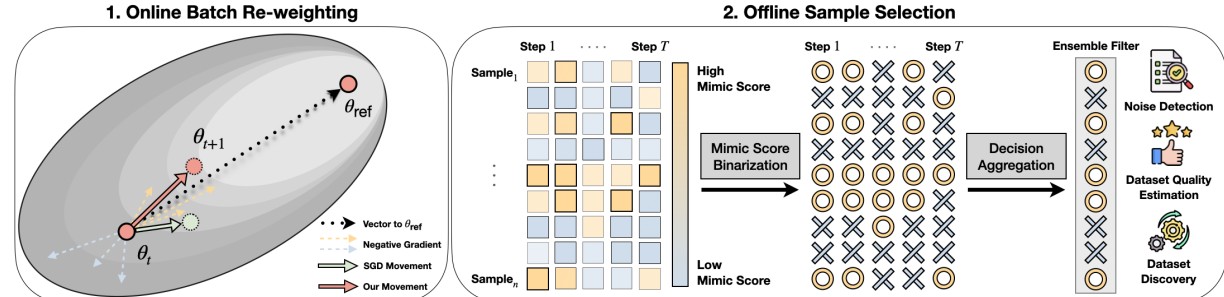

*Figure 2.* **Grad-Mimic Two-stage Workflow**: During training, Grad-Mimic measures the alignment between each sample's negative gradient and an induced vector from pre-trained reference model, then uses normalized alignment to re-weight gradients. After training, these alignment signals, mimic scores, are aggregated to identify low-value samples and used to construct an ensemble filter.

$v_t$, yielding an alignment score $m_{i,t}$, computed as follows

$$\text{Mimic\_Score}(s_{i,t}) := m_{i,t} = \frac{\langle -g_{i,t}, v_t \rangle}{\|v_t\|}. \quad (1)$$

This alignment score, named *Mimic Score*, reflects **how much a sample's directional compatibility can drive the model closer to a preferred weight space.** A sample having a lower mimic score suggests it has limited utility in guiding effective weight updates, making it a potential candidate for exclusion from future training.

## 4. Data Selection Framework

Building on this simple data-quality metric, we package it into Grad-Mimic, a two-stage data selection framework that first *prioritizes which samples to learn* (Sec. 4.1) and then *combines estimated utilities* to design *an effective ensemble data filter* (Sec. 4.2). Grad-Mimic workflow is in Figure 2.

### 4.1. Stage 1: Online Batch Re-weighting.

We first use mimic scores to aid data efficiency by *re-weighting*. Unlike standard gradient descent, which assigns uniform weights to all samples in a mini-batch, Grad-Mimic amplifies helpful samples and down-weights unhelpful ones.

To achieve this, each sample's mimic score is first normalized using the softmax function with a temperature parameter, $\tau$. The normalized score for a sample $s_{i,t}$ in a batch of size $b$ is computed as

$$\overline{m}_{i,t} = \frac{e^{m_{i,t}/\tau}}{\sum_{j=1}^{b} e^{m_{j,t}/\tau}}. \quad (2)$$

Then, the weight update step in Grad-Mimic is modified as

$$\theta_{t+1} := \theta_t - \eta \sum_{i=1}^{b} \overline{m}_{i,t} \cdot g_{i,t}. \quad (3)$$

The temperature $\tau$ controls the sensitivity of sample re-weighting, allowing us to adjust how sharply the model

prioritizes samples. A lower temperature results in a more aggressive focus on the most aligned samples, while a higher one encourages to converge to gradient descent.

**Theoretical Analysis.** We provide an analysis for *effective batch re-weighting* when training on a noisy dataset in Appendix A. In particular, we study conditions where Grad-Mimic's online method converges faster than standard gradient descent.

### 4.2. Stage 2: Offline Sample Selection.

The second stage of Grad-Mimic converts computed mimic scores—estimated sample utilities—into a unified assessment. This enables us to curate a higher-quality training dataset with a constructed data filter.

**Mimic Score Binarization.** Grad-Mimic first gathers normalized mimic scores from each sample at every training step. These indicate their intermediate contributions to learning. We convert these continuous scores into *binary retain/discard decisions* and explore three practical schemes:

- *Threshold-based*: A sample is chosen if its mimic score exceeds a defined threshold. A natural choice can be set as $1/b$ (indicating greater than uniform).
- *1D Clustering*: Samples are categorized into two groups using clustering (e.g., $k$-means (Wu, 1991) or Gaussian mixture), allowing unsupervised identification based on their assigned clusters.
- *Top-k Percent*: Samples are ranked by their mimic score, and the top-$k$ percent are chosen. The value of $k$ can be adjusted based on the available training budget.

**Decision Aggregation.** While binarized decisions provide instantaneous utility estimates at each training step, single-step assessments can be noisy (e.g., using early steps). Moreover, naive aggregation using majority vote may fail to capture learning dynamics. To address this, we treat each step's binarized assessments as weak votes and employ *weak supervision techniques* (Ratner et al., 2016; 2017; 2019; Fu et al., 2020) to combine filtering decisions across train-

ing steps. We start by learning a generative model to estimate the reliability of each step's assessments (Ratner et al., 2019). Once established, this model produces probabilistic aggregated decisions. This consensus can mitigate noise from inaccurate local gradients and models how a sample's utility evolves. Empirically, we employ the aggregation step from Snorkel framework (Ratner et al., 2017), a widely-adopted method in the weak supervision community.

**Broad Downstream Applications.** Ultimately, the combined filtering decisions yield a curated subset that can be used in subsequent training runs. In addition to dataset refinement, we can leverage their statistics to estimate dataset quality by either calculating *retention rate* (fraction of samples retained) or *averaged mimic score*. Moreover, the identified high-value samples provide data-quality insights *enable retrieval of similar samples to expand the current data pool*. Finally, Appendix G investigates mimic score's potential to infer training dataset used to produce a reference model. We discuss weak supervision setups and summarize algorithms in both online and offline stages in Appendix B.

# 5. Experiments

We evaluate Grad-Mimic's effectiveness using two experimental setups across diverse scales and domains. First, we test in a controlled setting, injecting label noise into standard datasets (Sec. 5.1). Then we evaluate on large-scale web-crawled datasets (Sec. 5.2). Our goals are to confirm key claims in both settings:

**C1. Effectiveness of Weight-Space Geometry.** Pre-trained model weights can act as a reliable reference for developing a new data-quality metric.

**C2. Enhanced Data Efficiency**: Prioritizing samples using mimic scores improves data efficiency and model performance, and accelerates convergence.

**C3. Minimal Computational Overhead**: Compared to prior model-based methods, mimic score computation incurs less overheads and reduced GPU memory usage.

**C4. Accurate Sample Identification**: Mimic scores precisely detect noisy samples, and their aggregated statistics highly correlate with overall dataset quality.

**C5. Effective Ensemble Filtering**: With limited sample assessments, Grad-Mimic can aggregate effective filtering decisions. The resulting ensemble filter complements existing data selection strategies.

## 5.1. Mislabeled Sample Detection

**Setups.** We begin with a controlled experiment by adding various levels of label noise to six image classification datasets. They are DTD (Cimpoi et al., 2014), Flowers102 (Nilsback & Zisserman, 2008), STL10 (Coates et al., 2011), OxfordIIIT Pet (Parkhi et al., 2012), CIFAR10, and CIFAR100 (Krizhevsky, 2009). We fine-tune a ViT-B/16 model (Dosovitskiy, 2020) on each noisy dataset under two training regimes: (i) *linear probing*, where only the final layer is tuned, and (ii) *full fine-tuning*, where gradients of all model parameters are re-weighted according to mimic scores. We normalize mimic scores with a temperature of 0.5 and use a batch size of 32. We simulate pre-trained reference models by training ViT-B/16 models on a noise-free split and use the *last-layer weights* as our reference to navigate training. We detail more training configurations in Appendix C.

After training, we binarize samples' mimic scores in two ways: setting one over batch size ($1/b$) as our threshold and using $k$-means and GMM to cluster. We then aggregate binarized outputs across training steps using Snorkel framework (Ratner et al., 2017).

**Expected Results.** We expect to validate that pre-trained reference model weights can serve as a reliable selection guide. Mimic scores can identify low-value mislabeled samples, improving data efficiency through online re-weighting.

**Baselines.** We compare Grad-Mimic's Stage 1 online method against six existing works that use gradient information for sample prioritization. We consider: (i) *Minibatch SGD*, no selection method, updates weights by averaging gradients within the mini-batch, (ii) *GraNd* (Paul et al., 2021) re-weights samples based on their gradient norm, prioritizing data that induce greater changes, (iii) *AGRA* (Sedova et al., 2023) computes the cosine similarity of each sample gradient to an average gradient from another random batch, then excludes outlier samples, (iv) *Grad-Match* (Killamsetty et al., 2021) adapts gradients by solving a subset selection problem to identify key samples within each batch, and (v) *influence function-based methods* (Wang et al., 2024a; Xia et al., 2024), approximating data influence by measuring gradient similarities with a noise-free validation dataset. Moreover, we include (vi) another reference model-based approach, *Rho-Loss* (Mindermann et al., 2022; Lin et al., 2024b), which uses a pre-trained model to derive excess losses and prioritizes training on worth learning ones.

**Stage 1 Results.** We report two stage results separately. For Stage 1, we evaluate online re-weighting effectiveness using testing accuracy under linear probing. Performance across all datasets and noise levels is shown in Table 1. Additional results under full fine-tuning are reported in Table 12 in Appendix H. We compare against influence function-based methods in Figure 3, where we introduce 50% label noise and vary the size of validation set used for computing sample influence. A detailed description of this experimental setup is offered in Appendix D. Furthermore, Appendix H investigates the stability of temperature $\tau$, the layer choice for mimicking, generalization to different reference models, scenario when facing new domains, and the impact of

*Table 1.* **Stage 1 Results in Mislabeled Sample Experiment**: Using mimic scores can effectively down-weight mislabeled samples during training, improving data efficiency and offering the greatest denoising capability.

| | DTD | | | Flowers102 | | | STL10 | | | Oxford-IIIT Pet | | | CIFAR10 | | | CIFAR100 | | | Average | | |
|---|---|---|---|---|---|---|---|---|---|---|---|---|---|---|---|---|---|---|---|---|---|
| Noise Level | 0.4 | 0.5 | 0.6 | 0.4 | 0.5 | 0.6 | 0.4 | 0.5 | 0.6 | 0.4 | 0.5 | 0.6 | 0.4 | 0.5 | 0.6 | 0.4 | 0.5 | 0.6 | 0.4 | 0.5 | 0.6 |
| SGD | 51.91 | 47.71 | 41.44 | 28.64 | 21.50 | 15.43 | 96.26 | 95.49 | 93.56 | 87.33 | 85.55 | 83.51 | 92.86 | 92.14 | 91.19 | 75.56 | 74.38 | 73.25 | 72.09 | 69.46 | 66.40 |
| Grad-Norm | 36.22 | 30.59 | 25.32 | 18.23 | 13.90 | 10.49 | 68.70 | 59.55 | 49.68 | 69.47 | 60.02 | 48.13 | 88.02 | 85.61 | 81.48 | 71.60 | 70.51 | 69.27 | 58.71 | 53.36 | 47.40 |
| AGRA | 41.81 | 36.28 | 31.49 | 41.81 | 36.28 | 31.49 | 96.19 | 95.15 | 93.11 | 85.25 | 82.53 | 78.39 | 92.51 | 92.01 | 90.99 | 72.50 | 71.30 | 69.69 | 71.68 | 68.93 | 65.86 |
| Grad-Match | 51.81 | 47.55 | 41.33 | 27.00 | 20.21 | 14.90 | 96.06 | 95.34 | 93.44 | 86.86 | 85.75 | 83.18 | 92.62 | 92.04 | 91.17 | 75.39 | 74.46 | 73.18 | 71.62 | 69.23 | 66.20 |
| Rho-Loss | 54.10 | 52.39 | 48.46 | 41.55 | 35.71 | 30.02 | 97.10 | 96.83 | 96.49 | 88.53 | 87.79 | 86.86 | 94.07 | 93.83 | **93.82** | 76.82 | 76.05 | 75.93 | 75.36 | 73.77 | 71.93 |
| **Grad-Mimic** | **54.68** | **54.10** | **50.43** | **42.75** | **37.10** | **31.71** | **97.16** | **97.00** | **96.90** | **88.80** | **88.25** | **87.14** | **94.15** | **93.92** | 93.80 | **77.24** | **76.82** | **76.05** | **75.80** | **74.53** | **73.01** |

*Table 2.* **Stage 2 Results in Mislabeled Sample Experiment**: We report detection results using F1-scores. Grad-Mimic accurately identifies mislabeled ones across datasets and binarization methods. It maintains strong performance under varying noise levels.

| | DTD | | | Flowers102 | | | STL10 | | | Oxford-IIIT Pet | | | CIFAR10 | | | CIFAR100 | | |
|---|---|---|---|---|---|---|---|---|---|---|---|---|---|---|---|---|---|---|
| Noise Level | 0.4 | 0.5 | 0.6 | 0.4 | 0.5 | 0.6 | 0.4 | 0.5 | 0.6 | 0.4 | 0.5 | 0.6 | 0.4 | 0.5 | 0.6 | 0.4 | 0.5 | 0.6 |
| Threshold (1/32) | 85.13 | 88.91 | 91.82 | 90.93 | 94.9 | **96.92** | 90.31 | 94.11 | **97.21** | 94.41 | 96.86 | **97.93** | 98.37 | **98.19** | 97.19 | 97.66 | **97.57** | 95.80 |
| 1D $k$-means | 77.13 | 76.68 | 77.17 | 84.45 | 87.62 | 87.47 | 78.90 | 83.04 | 78.82 | 95.80 | 95.49 | 93.73 | **98.62** | 98.17 | **97.56** | **98.10** | 97.46 | **96.62** |
| GMM | **97.85** | **96.72** | **95.68** | **98.39** | **96.96** | 94.21 | **97.96** | **97.16** | 95.69 | **98.04** | **97.07** | 95.86 | 95.70 | 92.89 | 88.62 | 95.86 | 94.31 | 92.55 |

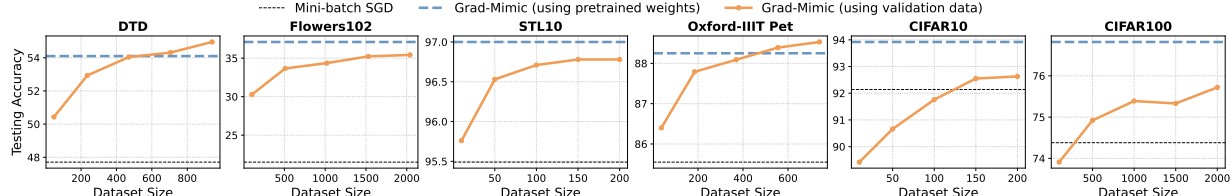

*Figure 3.* **Grad-Mimic outperforms influence function-based methods:** Leveraging the geometric information of the reference model as a selection guide provides several advantages, including *higher effectiveness, lower computational cost, and greater accessibility*.

reference model quality.

Results in Table 1 show that Grad-Mimic consistently achieves the highest average performance across different noise levels. This demonstrates its ability to ***more effectively identify and prioritize valuable samples, improving denoising capability***. In particular, Grad-Mimic outperforms reference model-based method Rho-Loss, which requires loading the full reference model and performing additional forward passes to compute excess loss. In contrast, Grad-Mimic evaluates sample utility directly from discrepancies in the last-layer model weights, a simpler matrix subtraction.

When compared against influence function-based approach, which approximates by using a golden validation set, Grad-Mimic achieves higher accuracy while avoiding their inherent limitations. These methods introduce dataset dependencies, incur additional gradient computation from validation set, and face challenges in ensuring its quality, as noted in GREATS (Wang et al., 2024a). For instance, on the Flowers102 and CIFAR100 dataset, achieving satisfactory results required over 2,000 correctly labeled samples—*yet still fell short of Grad-Mimic's performance*. These findings all validate our claims **C1 & C2**: *using the alignment with a target region in the weight space represented by a high-performing model can derive sample quality and enable better model training and improved data efficiency*.

**Efficiency Advantages.** In addition to the improved performance, Grad-Mimic offers significant efficiency benefits, ***including lower compute overheads, reduced GPU memory usage, and fewer steps to convergence***. We offer this efficiency analysis in Appendix E. Results support our claim **C3**, reducing computational overheads by a factor of 2.6.

**Stage 2 Results.** Next, we evaluate Grad-Mimic's ability to detect mislabeled samples by comparing aggregated predictions with ground-truth label noise and report F1-scores in Table 2. Grad-Mimic accurately identifies mislabeled samples across all datasets, ***achieving F1-scores above 95% and remaining robust to varying noise levels***. Additionally, Appendix F reports results when comparing to *majority vote* and using alternative aggregation methods, along with their runtime and data requirements. Results show that ***Stage 2 incurs negligible computational cost, and competitive performance can be achieved using only 0.5% of the data***.

Using CIFAR100 as an example, we visualize the distribution of mimic scores at each training step in Figure 4. In this dataset, half of the labels are flipped. The scores clearly separate into two groups, confirming that mimic scores provide informative signals for sample identification. Although the distributions vary over training, these dynamics are effectively captured by our aggregation step.

Finally, we use the ensemble filter from each dataset cre-

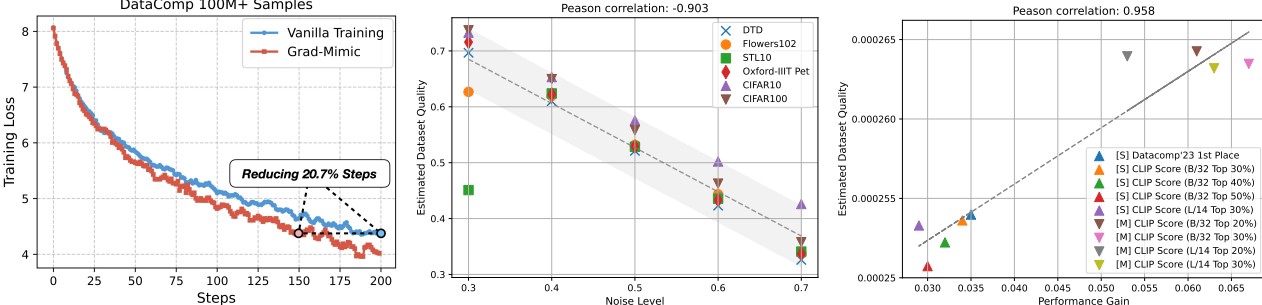

*Figure 4.* **Mimic Score Distribution**: Distributions extracted from CIFAR100 clearly separate by label correctness.

*Figure 5.* The left figure shows mimic score-guided training converges faster than vanilla baseline. The middle one uses retention rate to estimate training dataset quality under mislabeled sample experiments. The right figure takes the average on mimic scores in each curated dataset for performance gain estimation. [S/M] denotes the scale of DataComp dataset, small or medium, that the filter is designed on.

ated by GMM clustering to estimate training dataset quality based on the proportion of retained samples. The results, displayed in Figure 5 (middle), ***highly correspond to the presence of label noise, with a Pearson correlation of 0.903***. These demonstrate mimic score's effectiveness for sample identification and dataset quality estimation, validating our claims—**C4 & C5**.

### 5.2. Data Curation in Large-scale Web Datasets

**Setups.** We test Grad-Mimic in a more challenging setting using million-scale web-crawled datasets. We use the small- and medium-scale DataComp datasets (Gadre et al., 2023), which contain approximately 10 million and 100 million image-caption pairs, respectively, to train CLIP model from scratch (Radford et al., 2021). In these datasets, noise is *naturally* presented in the data. We follow the training setup from DataComp (Cherti et al., 2023; Gadre et al., 2023) and use publicly available pre-trained CLIP model's weights as our reference. This reference model, trained on the DataComp-1B dataset (1.4 billion samples), represents the best-performing model accessible to us at the time of experimentation. It serves as a proxy for the ideal reference point for scenarios *where training such powerful models is infeasible*. We use the final MLP layer in the text and image encoders respectively as our target. We evaluate resulting model performance via DataComp benchmark, which includes 38 diverse image classification and retrieval tasks. Appendix C provides complete implementation details.

In Stage 2, we use top-$k\%$ approach to binarize scores and create the ensemble filter, subsequently training a new CLIP model on the curated dataset to evaluate constructed filter.

*Table 3.* **Stage 1 Results in DataComp Experiment**: On both scales, with the aid of publicly available pre-trained weights, Grad-Mimic yields higher CLIP model performance. Full table with various temperature settings is placed in Table 13 in Appendix H.

| Scale | Training Method | Mimic Layer $\theta_{\text{ref}}$ | Temperature $\tau$ | Average over 38 datasets (↑) |
|---|---|---|---|---|
| Small | Vanilla Training | — | — | 0.131 |
| | **Grad-Mimic** | Last MLP Layer in Text Encoder | 0.05 | **0.135** |
| | | | 0.5 | **0.139** |
| | | Last MLP Layer in Image Encoder | 0.05 | **0.146** |
| | | | 0.5 | **0.145** |
| Medium | Vanilla Training | — | — | 0.254 |
| | **Grad-Mimic** | Last MLP Layer in Image Encoder | 0.05 | **0.258** |

**Expected Results.** We expect mimic scores help large-scale training focus on high-value samples and can serve as an instrumental metric for automating effective data curation.

**Baselines.** For the first stage, we compare Grad-Mimic to vanilla training, where each sample contributes in the weight update equally. For Stage 2 comparison, we study our mimic score-based filter with the following methods: (i) *No Filtering*: using the entire training pool, (ii) *Basic Filtering* (Gadre et al., 2023): selecting samples based on predefined criteria like caption length, image size, and caption language—representing a *human-designed filter*, (iii) *CLIP Score* (Hessel et al., 2021): selecting top-$k\%$ samples based on embedding similarity between images and captions. Scores are computed by OpenAI's CLIP ViT-B/32 model (Radford et al., 2021).

**Stage 1 Results.** Table 3 presents the pretraining results for CLIP models on both dataset scales. Full results using

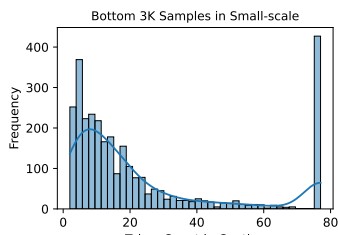 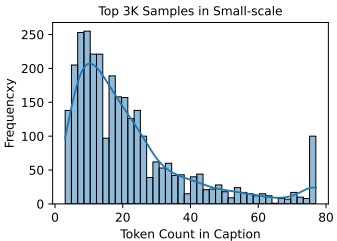 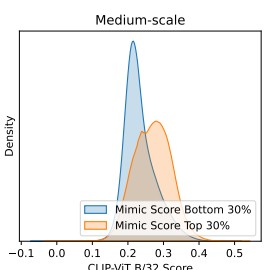 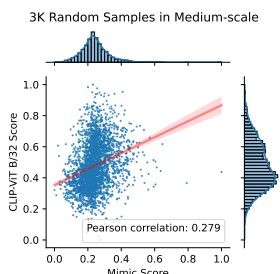

*Figure 6.* The left two figures compare token counts using bottom and top samples ranked by mimic score. This filtering logic matches handcrafted heuristics, where low-value samples have captions that are either *too short* or *over the 77-token limit*. The right two figures show mimic score distribution and its correlation with CLIP score.

*Table 4.* **Stage 2 Results in DataComp Experiment**: Mimic score-based filters augment CLIP score-based filters by removing low-value samples. Symbol "\" denotes the exclusion of curated datasets.

| Scale | Filtering Strategy | Training Dataset Size | Average over 38 datasets (↑) |
|---|---|---|---|
| Small | No Filtering | 10.7M | 0.131 |
| | CLIP Score (B/32 Top 30%) | 3M | 0.165 |
| | CLIP Score \ Mimic Score Bottom 15% | **2.7M (0.3M ↓)** | **0.163** |
| | CLIP Score \ Mimic Score Bottom 20% | **2.6M (0.4M ↓)** | **0.164** |
| Medium | No Filtering | 101.9M | 0.254 |
| | CLIP Score (B/32 Top 30%) | 30M | 0.321 |
| | CLIP Score \ Mimic Score Bottom 5% | **28.1M (1.9M ↓)** | **0.323** |
| | CLIP Score \ Mimic Score Bottom 10% | **27.7M (2.3M ↓)** | **0.323** |
| | CLIP Score \ Mimic Score Bottom 30% | **25.3M (4.7M ↓)** | **0.322** |

various temperatures are provided in Appendix H. Grad-Mimic consistently outperforms standard training across all temperature settings. Interestingly, we find that mimicking the last layer of the image encoder yields more performance gains compared to targeting the text encoder. Furthermore, Figure 5 (left) presents the learning curves for CLIP models trained with Grad-Mimic versus the vanilla method. *Grad-Mimic achieves faster convergence while producing better CLIP models, using 20.7% fewer steps*. These findings on the scale-up testbed further support our claims **C1 & C2**.

**Stage 2 Results.** We use the derived mimic scores from our best-performing model in Stage 1 to design filters and evaluate their effectiveness. We study the broader applicability of mimic scores as a complement signal when combined with existing filters. We remove low-value samples ranked by mimic score to augment a CLIP score-based filter (B/32 Top 30%). Results are presented in Table 4. This complementary approach not only enhances model performance but also improves data efficiency by reducing training set size, *specifically removing 4.7 million samples in the medium-scale dataset.* We further compare our developed filters against basic filtering strategies, confirming Grad-Mimic effectively automates selection process, outperforming human-designed one in Appendix H.

Finally, we gather mimic scores from datasets that are curated by various CLIP score-based filters. We include the top-ranked approach during our time of experimen-

tation (Huang et al., 2024b), which uses an orthogonal approach based on object detection models. We average samples' mimic scores in each curated dataset as dataset quality estimates and compare with their corresponding performance gains. The results, shown in Figure 5 (right), reveal a positive alignment with a Pearson correlation of **0.958**, demonstrating the feasibility to *use mimic score as a reliable metric to predict model performance gains*. These again support our claims, **C4 & C5**.

**Sample Analysis.** We analyze high- and low-value samples ranked by mimic score. In Figure 6, we compare token count in the web-crawled caption. We find that bottom-ranked samples often carry captions that are either too short (1-2 words) or excessively long (over the maximum token limit), unlike top-ranked samples. Moreover, mimic score-based filter effectively identifies low-value samples, such as misaligned-caption images in the medium-scale dataset (see Figure 1). *These filtering patterns align with human intuition but are automatically captured by Grad-Mimic.*

We visualize the top and bottom 30% of samples along with their CLIP scores in Figure 6. High-value samples align with higher CLIP scores, while low-value ones are more concentrated in the lower range (half below 0.2). Furthermore, we calculate their correlation on random samples. We normalize both scores into the range 0 to 1 for clarity. The positive correlation supports our claim **C4 & C5**, mimic score is effective at identifying high-quality samples, as reflected by high CLIP scores.

## 6. Conclusion

We introduce Mimic Score, an efficient data-quality metric that assesses sample utility by exploiting the weight-space geometry from a pre-trained model and the alignment using sample gradient. Leveraging this metric, we propose Grad-Mimic, a two-stage framework including online prioritization and offline selection. Empirically, Grad-Mimic improves data efficiency, accelerates convergence, accurately detects mislabeled samples, and enhances existing filters, all while incurring minimal computational overheads.

# 7. Impact Statements

Data selection is a crucial prerequisite for training modern models. In this work, we demonstrate that Grad-Mimic effectively identifies and removes low-value samples, enhancing data efficiency and accelerating training. We do not foresee any direct negative social consequences from the technique itself.

However, if the reference model used to evaluate sample utility is poorly calibrated or biased, there remains a risk of refining low-quality datasets. Appendix H investigates such scenario. We study the sensitivity to different reference models and the impact of reference weight quality. Our results indicate that Grad-Mimic is remarkably robust: performance remains stable when targeting different regions of the weight-space (see Table 17). Additionally, performance remains competitive even when the reference model is heavily degraded by noise (see Figure 10), or trained on limited resources (using 372× smaller dataset in Table 18).

In practice, we believe this *reference model dependency* can be easily mitigated by validating the reference model on a small, trusted benchmark prior to sample scoring. This lightweight check also applies when choosing among multiple candidate reference models. Compared to many recent data selection approaches that depend on high-quality validation sets, ***ensuring reasonable reference model quality is substantially easier***. Overall, Grad-Mimic provides greater flexibility and control by decoupling data scoring from such *hard-to-obtain* resources.

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

**Appendix Roadmap.** Our appendix is structured as follows. It starts with a theoretical analysis of effective batch re-weighting in Appendix A, studying conditions where Grad-Mimic's online re-weighting method outperforms standard gradient descent. Next, Appendix B outlines Grad-Mimic algorithms in both stages. This leads into detailed experimental setups, including training configurations and computational resources, presented in Appendix C. Then, we continue with Appendix D, providing further discussion when comparing against other model-based data selection approaches. After this, Appendix E provides a comprehensive analysis about Grad-Mimic's efficiency advantages, substantiated by various evidence. We then incorporate additional aggregation alternatives from the weak supervision literature and study their resulting performance, needed runtime, and data requirement in Appendix F. We further present an interesting application of mimic score for discovering training samples used in reference models in Appendix G. Appendix H wraps up with extensive ablation studies. We investigate temperature stability, extension to text-domain tasks, choices of layer weights, generalization to difference reference models, the impact of reference model quality, scenario when facing new domains, and comparison to human-designed filters. Finally, we discuss potential directions in our future work in Appendix I.

## A. Theoretical Analysis

We first present a glossary table listing the defined notations and their meanings used in the following analysis.

| Symbol | Meaning |
|---|---|
| $\mathbb{E}$ | Expectation over inherent randomness, such as over gradient noise $\delta$. |
| $\mathbb{E}^{(x)}$ | Expectation over the dataset |
| $\sigma(\cdot)$ | The softmax function |
| $n$ | Dataset size |
| $L$ | Smoothness parameter on loss function $\ell$ |
| $\delta$ | Level of the gradient noise, $\delta = \mathbb{E}[\|\delta_{i,t}\|^2]$ |
| $\epsilon$ | Parameter distance between reference model and the optimal model, $\epsilon = \|\theta_{\text{ref}} - \theta_*\|$ |
| $\tau$ | Temperature parameter used in Grad-Mimic updates |
| $v_t$ | Target vector induced by reference model, $\theta_{\text{ref}} - \theta_t$ |
| $-a_{i,t}$ | The alignment of the gradient of the loss with the target vector, $g_{i,t}^\top v_t$ |
| $\lambda_t$ | The softmax parameter, $1/(\tau\|v_t\|)$ |
| $\ell(\theta_t; s_i)$ | The loss of the model with parameters $\theta_t$ on a given sample $s_i$ |
| $g_{i,t}$ | $\nabla\ell(\theta_t; s_i)$ |
| $\tilde{g}_{i,t}$ | $\nabla\ell(\theta_t; s_i) + \delta_{i,t}$ where $\delta_{i,t}$ is a noise term |
| $\theta_{t+1}^{\text{gd}}$ | The learned parameter through a standard GD update |
| $\theta_{t+1}^{\text{gm}}$ | The learned parameter through a Grad-Mimic update |

We aim to analyze the convergence rates of standard Gradient Descent (GD) and the Grad-Mimic (GM) online re-weighting algorithm for an $L$-smooth loss function $\ell$, using noisy gradients with noise level $\delta$. Our goal is to compare their convergence to the global minimum $\theta_*$, explicitly incorporating the term $\epsilon = \|\theta_{\text{ref}} - \theta_*\|$ for Grad-Mimic. We start by specifying the update rules for both algorithms, then present three lemmas concerning adaptive algorithms and the softmax function. Finally, we discuss how these results apply to Grad-Mimic and quantify the improvement over GD.

**Problem Setup.** Consider a (potentially noisy) dataset $D = \{s_i\}_{i=1}^n$ and an $L$-smooth loss function $\ell(\theta; s)$, defined as:

$$\ell(\theta) = \frac{1}{n}\sum_{i=1}^n \ell(\theta; s_i).$$

The function $\ell$ is $L$-smooth, and its global minimum is denoted by $\theta_*$. We assume access to noisy gradients:

$$\tilde{g}_{i,t} = g_{i,t} + \delta_{i,t},$$

where the gradient noise $\delta_{i,t}$ satisfies:

$$\mathbb{E}[\delta_{i,t}] = 0, \quad \mathbb{E}[\|\delta_{i,t}\|^2] = \delta.$$

Additionally, we have a reference solution $\theta_{\text{ref}}$ closer to the optimum than the initial weight $\theta_0$ in the weight space:

$$\|\theta_0 - \theta_*\| \gg \|\theta_{\text{ref}} - \theta_*\| = \epsilon.$$

**Standard Gradient Descent.** An update rule for GD at iteration $t$ is:

$$\theta_{t+1}^{\text{gd}} = \theta_t - \eta \frac{1}{n} \sum_{i=1}^{n} \tilde{g}_{i,t}. \tag{4}$$

**Grad-Mimic's Online Re-weighting.** An update rule for Grad-Mimic at iteration $t$ is:

$$\theta_{t+1}^{\text{gm}} = \theta_t - \frac{\eta}{\sum_{i=1}^{n} \alpha_{i,t}} \sum_{i=1}^{n} \alpha_{i,t} \tilde{g}_{i,t}, \tag{5}$$

with adaptive weighting that:

$$v_t = \theta_{\text{ref}} - \theta_t,$$
$$m_{i,t} = -\tilde{g}_{i,t}^{\top} v_t / \|v_t\|,$$
$$\alpha_{i,t} = e^{m_{i,t}/\tau}.$$

This adaptive weighting leverages the proximity of the reference point $\theta_{\text{ref}}$ to the true minimum $\theta_*$, which is expected to effectively mitigating gradient noise. When it is convenient, the following notation is used:

$$a_{i,t} = -\tilde{g}_{i,t}^{\top} v_t,$$
$$\lambda = 1/(\tau \|v_t\|),$$

so that

$$\alpha_{i,t} = \exp(\lambda a_{i,t}).$$

We begin with a lemma about *any adaptive weighting procedure*, where the $\alpha_{i,t}$ (written as $\alpha_i$ unless the timestep needs to be specified) terms are unconstrained.

**Lemma A.1.** *Let*

$$R = 2Cov^{(x)}\left(\frac{\alpha_i}{\mathbb{E}^{(x)}[\alpha_i]}, a_{i,t}\right) - \eta\left(\|\mathbb{E}^{(x)}[\frac{\alpha_i}{\mathbb{E}^{(x)}[\alpha_i]} \tilde{g}_{i,t}]\|^2 - \|\mathbb{E}^{(x)}[\tilde{g}_{i,t}]\|^2\right)$$

*Then*

$$\|\theta_{t+1}^{gm} - \theta_{ref}\|^2 = \|\theta_{t+1}^{gd} - \theta_{ref}\|^2 - \eta R$$

*Proof.* First, begin with a calculation of what $\|\theta_{t+1}^{\text{gm}} - \theta_{\text{ref}}\|^2$ is.

$$\|\theta_{t+1}^{\text{gm}} - \theta_{\text{ref}}\|^2 = \|\theta_t - \eta \frac{\sum_{i=1}^{n} \alpha_i \tilde{g}_{i,t}}{\sum_{i=1}^{n} \alpha_i} - \theta_{\text{ref}}\|^2$$

$$= \|\theta_t - \eta \frac{\mathbb{E}^{(x)}[\alpha_i \tilde{g}_{i,t}]}{\mathbb{E}^{(x)}[\alpha_i]} - \theta_{\text{ref}}\|^2$$

$$= \|\theta_t - \theta_{\text{ref}}\|^2 - 2\eta(\theta_t - \theta_{\text{ref}})^{\top} \frac{\mathbb{E}^{(x)}[\alpha_i \tilde{g}_{i,t}]}{\mathbb{E}^{(x)}[\alpha_i]} + \eta^2 \left\|\frac{\mathbb{E}^{(x)}[\alpha_i \tilde{g}_{i,t}]}{\mathbb{E}^{(x)}[\alpha_i]}\right\|^2$$

$$= \|\theta_t - \theta_{\text{ref}}\|^2 + 2\eta \frac{\mathbb{E}^{(x)}[\alpha_i v_t^{\top} \tilde{g}_{i,t}]}{\mathbb{E}^{(x)}[\alpha_i]} + \eta^2 \frac{\|\mathbb{E}^{(x)}[\alpha_i \tilde{g}_{i,t}]\|^2}{\mathbb{E}^{(x)}[\alpha_i]^2}$$

Next, using the following identities,

$$\frac{\mathbb{E}^{(x)}[\alpha_i v_t^{\top} \tilde{g}_{i,t}]}{\mathbb{E}^{(x)}[\alpha_i]} = \frac{\mathbb{E}^{(x)}[\alpha_i v_t^{\top} \tilde{g}_{i,t}] - \mathbb{E}^{(x)}[\alpha_i]\mathbb{E}^{(x)}[v_t^{\top} \tilde{g}_{i,t}] + \mathbb{E}^{(x)}[\alpha_i]\mathbb{E}^{(x)}[v_t^{\top} \tilde{g}_{i,t}]}{\mathbb{E}^{(x)}[\alpha_i]}$$

$$= \mathbb{E}^{(x)}[v_t^{\top} \tilde{g}_{i,t}] + \frac{\mathbb{E}^{(x)}[\alpha_i v_t^{\top} \tilde{g}_{i,t}] - \mathbb{E}^{(x)}[\alpha_i]\mathbb{E}^{(x)}[v_t^{\top} \tilde{g}_{i,t}]}{\mathbb{E}^{(x)}[\alpha_i]}$$

$$= \mathbb{E}^{(x)}[v_t^{\top} \tilde{g}_{i,t}] + \frac{Cov^{(x)}(\alpha_i, v_t^{\top} \tilde{g}_{i,t})}{\mathbb{E}^{(x)}[\alpha_i]}$$

and

$$\frac{\|\mathbb{E}^{(x)}[\alpha_i \tilde{g}_{i,t}]\|^2}{\mathbb{E}^{(x)}[\alpha_i]^2} = \frac{\|\mathbb{E}^{(x)}[\alpha_i \tilde{g}_{i,t}]\|^2 - \mathbb{E}^{(x)}[\alpha_i]^2 \|\mathbb{E}^{(x)}[\tilde{g}_{i,t}]\|^2 + \mathbb{E}^{(x)}[\alpha_i]^2 \|\mathbb{E}^{(x)}[\tilde{g}_{i,t}]\|^2}{\mathbb{E}^{(x)}[\alpha_i]^2}$$

$$= \|\mathbb{E}^{(x)}[\tilde{g}_{i,t}]\|^2 + \frac{\|\mathbb{E}^{(x)}[\alpha_i \tilde{g}_{i,t}]\|^2 - \mathbb{E}^{(x)}[\alpha_i]^2 \|\mathbb{E}^{(x)}[\tilde{g}_{i,t}]\|^2}{\mathbb{E}^{(x)}[\alpha_i]^2}$$

$$= \|\mathbb{E}^{(x)}[\tilde{g}_{i,t}]\|^2 + \frac{\|\mathbb{E}^{(x)}[\alpha_i \tilde{g}_{i,t}]\|^2 - \|\mathbb{E}^{(x)}[\alpha_i]\mathbb{E}^{(x)}[\tilde{g}_{i,t}]\|^2}{\mathbb{E}^{(x)}[\alpha_i]^2}$$

We can proceed as

$$\|\theta_{t+1}^{\text{gm}} - \theta_{\text{ref}}\|^2 = \|\theta_t - \theta_{\text{ref}}\|^2 + 2\eta \left( \mathbb{E}^{(x)}[v_t^\top \tilde{g}_{i,t}] + \frac{\text{Cov}^{(x)}(\alpha_i, v_t^\top \tilde{g}_{i,t})}{\mathbb{E}^{(x)}[\alpha_i]} \right)$$

$$+ \eta^2 \left( \|\mathbb{E}^{(x)}[\tilde{g}_{i,t}]\|^2 + \frac{\|\mathbb{E}^{(x)}[\alpha_i \tilde{g}_{i,t}]\|^2 - \|\mathbb{E}^{(x)}[\alpha_i]\mathbb{E}^{(x)}[\tilde{g}_{i,t}]\|^2}{\mathbb{E}^{(x)}[\alpha_i]^2} \right)$$

$$= \|\theta_t - \theta_{\text{ref}}\|^2 + 2\eta \mathbb{E}^{(x)}[v_t^\top \tilde{g}_{i,t}] + \eta^2 \|\mathbb{E}^{(x)}[\tilde{g}_{i,t}]\|^2$$

$$+ 2\eta \frac{\text{Cov}^{(x)}(\alpha_i, v_t^\top \tilde{g}_{i,t})}{\mathbb{E}^{(x)}[\alpha_i]} + \eta^2 \frac{\|\mathbb{E}^{(x)}[\alpha_i \tilde{g}_{i,t}]\|^2 - \|\mathbb{E}^{(x)}[\alpha_i]\mathbb{E}^{(x)}[\tilde{g}_{i,t}]\|^2}{\mathbb{E}^{(x)}[\alpha_i]^2}$$

$$= \|\theta_{t+1}^{\text{gd}} - \theta_{\text{ref}}\|^2 + 2\eta \frac{\text{Cov}^{(x)}(\alpha_i, v_t^\top \tilde{g}_{i,t})}{\mathbb{E}^{(x)}[\alpha_i]} + \eta^2 \frac{\|\mathbb{E}^{(x)}[\alpha_i \tilde{g}_{i,t}]\|^2 - \|\mathbb{E}^{(x)}[\alpha_i]\mathbb{E}^{(x)}[\tilde{g}_{i,t}]\|^2}{\mathbb{E}^{(x)}[\alpha_i]^2}$$

$$= \|\theta_{t+1}^{\text{gd}} - \theta_{\text{ref}}\|^2 + 2\eta \text{Cov}^{(x)}(\frac{\alpha_i}{\mathbb{E}^{(x)}[\alpha_i]}, v_t^\top \tilde{g}_{i,t}) + \eta^2 (\|\mathbb{E}^{(x)}[\frac{\alpha_i}{\mathbb{E}^{(x)}[\alpha_i]}\tilde{g}_{i,t}]\|^2 - \|\mathbb{E}^{(x)}[\tilde{g}_{i,t}]\|^2)$$

$$= \|\theta_{t+1}^{\text{gd}} - \theta_{\text{ref}}\|^2 - 2\eta \text{Cov}^{(x)}(\frac{\alpha_i}{\mathbb{E}^{(x)}[\alpha_i]}, a_{i,t}) + \eta^2 (\|\mathbb{E}^{(x)}[\frac{\alpha_i}{\mathbb{E}^{(x)}[\alpha_i]}\tilde{g}_{i,t}]\|^2 - \|\mathbb{E}^{(x)}[\tilde{g}_{i,t}]\|^2)$$

$$= \|\theta_{t+1}^{\text{gd}} - \theta_{\text{ref}}\|^2 - \eta R$$

$\square$

From Lemma A.1, we want to bound $R$ to hopefully show that $R > 0$, indicating that taking an update step with Grad-Mimic will converge faster than GD.

**Lemma A.2.** *Let* $f(\lambda) = \frac{\sum_{i=1}^n \exp(\lambda v_i) v_i}{\sum_{i=1}^n \exp(\lambda v_i)}$ *and* $\sigma(\lambda)$ *be the vector where* $\sigma(\lambda)_i = \frac{\exp(\lambda v_i)}{\sum_{j=1}^n \exp(\lambda v_j)}$. *Then*

$$f(\lambda) - f(0) \geq \frac{1}{\lambda} \|\sigma(\lambda) - \sigma(0)\|^2$$

*Proof.* Following the definitions,

$$f(\lambda) - f(0) = \sigma(\lambda)^\top v - \sigma(0)^\top v$$
$$= (\sigma(\lambda) - \sigma(0))^\top (v - 0)$$

Now, fix a $\lambda$. Abusing notation, let $\sigma(v)_i = \frac{\exp(\lambda v_i)}{\sum_{j=1}^n \exp(\lambda v_j)}$, where $\lambda$ is now fixed and the vector $v$ is variable. As such, $f(\lambda) - f(0) = (\sigma(v) - \sigma(0))^\top (v - 0)$. Since $\sigma(\cdot)$ is $\lambda$-Lipschitz, it follows that

$$(\sigma(v) - \sigma(0))^\top (v - 0) \geq \frac{1}{\lambda} \|\sigma(v) - \sigma(0)\|^2$$

then we can have

$$f(\lambda) - f(0) \geq \frac{1}{\lambda} \|\sigma(\lambda) - \sigma(0)\|^2$$

$\square$

**Lemma A.3.** *The value of $R$ in Grad-Mimic is guaranteed to be positive if*

$$\eta < \frac{2\tau\|v_t\| \cdot \|\sigma(1/\tau\|v_t\|) - \sigma(0)\|^2}{\|\mathbb{E}^{(x)}[\frac{\alpha_i}{\mathbb{E}^{(x)}[\alpha_i]}\tilde{g}(\theta_t; s_i)]\|^2 - \|\mathbb{E}^{(x)}[\tilde{g}(\theta_t; s_i)]\|^2},$$

*or $\|\mathbb{E}^{(x)}[\frac{\alpha_i}{\mathbb{E}^{(x)}[\alpha_i]}\tilde{g}(\theta_t; s_i)]\|^2 \leq \|\mathbb{E}^{(x)}[\tilde{g}(\theta_t; s_i)]\|^2$.*

*Proof.* First, recall

$$R = 2\text{Cov}^{(x)}(\frac{\alpha_i}{\mathbb{E}^{(x)}[\alpha_i]}, (\theta_t - \theta_{\text{ref}})^\top \tilde{g}(\theta_t; s_i)) - \eta(\|\mathbb{E}^{(x)}[\frac{\alpha_i}{\mathbb{E}^{(x)}[\alpha_i]}\tilde{g}(\theta_t; s_i)]\|^2 - \|\mathbb{E}^{(x)}[\tilde{g}(\theta_t; s_i)]\|^2)$$

Inspecting the first term, as a result of the previous lemma,

$$\begin{aligned}
\text{Cov}^{(x)}(\frac{\alpha_i}{\mathbb{E}^{(x)}[\alpha_i]}, (\theta_t - \theta_{\text{ref}})^\top \tilde{g}(\theta_t; s_i)) &= \text{Cov}^{(x)}(\frac{\alpha_i}{\mathbb{E}^{(x)}[\alpha_i]}, \tilde{a}_{i,t}) \\
&= \mathbb{E}^{(x)}[\frac{\alpha_i}{\mathbb{E}^{(x)}[\alpha_i]}\tilde{a}_{i,t}] - \mathbb{E}^{(x)}[\tilde{a}_{i,t}] \\
&= \frac{\sum_{i=1}^n \alpha_i \tilde{a}_{i,t}}{\sum_{i=1}^n \alpha_i} - \frac{\sum_{i=1}^n \exp(0)\tilde{a}_{i,t}}{\sum_{i=1}^n \exp(0)} \\
&= \frac{\sum_{i=1}^n \exp(\lambda\tilde{a}_{i,t})\tilde{a}_{i,t}}{\sum_{i=1}^n \exp(\lambda\tilde{a}_{i,t})} - \frac{\sum_{i=1}^n \exp(0\tilde{a}_{i,t})\tilde{a}_{i,t}}{\sum_{i=1}^n \exp(0\tilde{a}_{i,t})} \\
&= f(\lambda) - f(0) \\
&\geq \frac{1}{\lambda}\|\sigma(\lambda) - \sigma(0)\|^2
\end{aligned}$$

Thus,

$$R \geq \frac{2}{\lambda}\|\sigma(\lambda) - \sigma(0)\|^2 - \eta(\|\mathbb{E}^{(x)}[\frac{\alpha_i}{\mathbb{E}^{(x)}[\alpha_i]}\tilde{g}(\theta_t; s_i)]\|^2 - \|\mathbb{E}^{(x)}[\tilde{g}(\theta_t; s_i)]\|^2)$$

Finally, with the definition $\lambda = 1/(\tau\|v_t\|)$,

$$R \geq 2\tau\|v_t\| \cdot \|\sigma(1/\tau\|v_t\|) - \sigma(0)\|^2 - \eta(\|\mathbb{E}^{(x)}[\frac{\alpha_i}{\mathbb{E}^{(x)}[\alpha_i]}\tilde{g}(\theta_t; s_i)]\|^2 - \|\mathbb{E}^{(x)}[\tilde{g}(\theta_t; s_i)]\|^2)$$

A sufficient condition for $R > 0$ is therefore given by

$$\eta < \frac{2\tau\|v_t\| \cdot \|\sigma(1/\tau\|v_t\|) - \sigma(0)\|^2}{\|\mathbb{E}^{(x)}[\frac{\alpha_i}{\mathbb{E}^{(x)}[\alpha_i]}\tilde{g}(\theta_t; s_i)]\|^2 - \|\mathbb{E}^{(x)}[\tilde{g}(\theta_t; s_i)]\|^2}$$

$\square$

So far, the only quantity being bounded is the distance to the reference model $\theta_{\text{ref}}$. Lastly, we extend to bound convergence to the optimum $\theta_*$ and present our Theorem A.4 by using 3 lemma above.

**Theorem A.4.** *Let $\kappa = \max_i \|\tilde{g}(\theta_t; s_i)\|$. Then*

$$\|\theta_{t+1}^{gm} - \theta_*\|^2 \leq \|\theta_{t+1}^{gd} - \theta_*\|^2 - \eta\left[2\tau\|v_t\| \cdot \|\sigma(1/\tau\|v_t\|) - \sigma(0)\|^2 - 4\epsilon\kappa - \eta\kappa^2\right]$$

*Specifically, if $\eta < \frac{2\tau\|v_t\| \cdot \|\sigma(1/\tau\|v_t\|) - \sigma(0)\|^2 - 4\epsilon\kappa}{\kappa^2}$ then $\|\theta_{t+1}^{gm} - \theta_*\| \leq \|\theta_{t+1}^{gd} - \theta_*\|$.*

*Proof.* Starting with an application of Lemma A.1,

$$\begin{aligned}
\|\theta_{t+1}^{gm} - \theta_*\|^2 &= \|\theta_{t+1}^{gm} - \theta_{\text{ref}}\|^2 + \|\theta_{\text{ref}} - \theta_*\|^2 + 2(\theta_{t+1}^{gm} - \theta_{\text{ref}})^\top(\theta_{\text{ref}} - \theta_*) \\
&= \|\theta_{t+1}^{gd} - \theta_{\text{ref}}\|^2 + \|\theta_{\text{ref}} - \theta_*\|^2 + 2(\theta_{t+1}^{gm} - \theta_{\text{ref}})^\top(\theta_{\text{ref}} - \theta_*) - \eta R \\
&= \|\theta_{t+1}^{gd} - \theta_*\|^2 - 2(\theta_{t+1}^{gd} - \theta_{\text{ref}})^\top(\theta_{\text{ref}} - \theta_*) + 2(\theta_{t+1}^{gm} - \theta_{\text{ref}})^\top(\theta_{\text{ref}} - \theta_*) - \eta R \\
&= \|\theta_{t+1}^{gd} - \theta_*\|^2 + 2(\theta_{t+1}^{gm} - \theta_{t+1}^{gd})^\top(\theta_{\text{ref}} - \theta_*) - \eta R \\
&\leq \|\theta_{t+1}^{gd} - \theta_*\|^2 + 2\epsilon\|\theta_{t+1}^{gd} - \theta_{t+1}^{gm}\| - \eta R
\end{aligned}$$

Thus, we achieve superior performance towards the optimum if

$$R > \frac{2\epsilon \|\theta_{t+1}^{\text{gm}} - \theta_{t+1}^{\text{gd}}\|}{\eta}$$

Using Lemma A.3,

$$
\begin{aligned}
\|\theta_{t+1}^{\text{gm}} - \theta_*\|^2 &\le \|\theta_{t+1}^{\text{gd}} - \theta_*\|^2 + 2\epsilon\|\theta_{t+1}^{\text{gd}} - \theta_{t+1}^{\text{gm}}\| - \eta R \\
&\le \|\theta_{t+1}^{\text{gd}} - \theta_*\|^2 + 2\epsilon\|\theta_{t+1}^{\text{gd}} - \theta_{t+1}^{\text{gm}}\| - 2\eta\tau\|v_t\| \cdot \|\sigma(1/\tau\|v_t\|) - \sigma(0)\|^2 \\
&\quad + \eta^2(\|\mathbb{E}^{(x)}[\frac{\alpha_i}{\mathbb{E}^{(x)}[\alpha_i]}\tilde{g}(\theta_t; s_i)]\|^2 - \|\mathbb{E}^{(x)}[\tilde{g}(\theta_t; s_i)]\|^2) \\
&\le \|\theta_{t+1}^{\text{gd}} - \theta_*\|^2 + 4\epsilon\eta\kappa - 2\eta\tau\|v_t\| \cdot \|\sigma(1/\tau\|v_t\|) - \sigma(0)\|^2 \\
&\quad + \eta^2(\|\mathbb{E}^{(x)}[\frac{\alpha_i}{\mathbb{E}^{(x)}[\alpha_i]}\tilde{g}(\theta_t; s_i)]\|^2 - \|\mathbb{E}^{(x)}[\tilde{g}(\theta_t; s_i)]\|^2) \\
&\le \|\theta_{t+1}^{\text{gd}} - \theta_*\|^2 + 4\epsilon\eta\kappa - 2\eta\tau\|v_t\| \cdot \|\sigma(1/\tau\|v_t\|) - \sigma(0)\|^2 + \eta^2\kappa^2
\end{aligned}
$$

where the penultimate inequality uses $\|\theta_{t+1}^{\text{gm}} - \theta_{t+1}^{\text{gd}}\| \le \|\theta_{t+1}^{\text{gm}}\| + \|\theta_{t+1}^{\text{gd}}\| \le 2\eta\kappa$.

We then can find a bound of $\eta$ for $\|\theta_{t+1}^{\text{gm}} - \theta_*\|^2 \le \|\theta_{t+1}^{\text{gd}} - \theta_*\|^2$:

$$\eta < \frac{2\tau\|v_t\| \cdot \|\sigma(1/\tau\|v_t\|) - \sigma(0)\|^2 - 4\epsilon\kappa}{\kappa^2}.$$

$\square$

**Theorem A.4 Discussion.** Our convergence bound reveals conditions under which Grad-Mimic outperforms GD. First, the learning rate $\eta$ should be small. A large $\eta$ causes the $\kappa^2$ term to dominate, leading to large, erroneous steps in Grad-Mimic compared to GD. This is particularly evident when the temperature is small, as Grad-Mimic amplifies specific gradients while GD averages conflicting ones, potentially overshooting the optimum if the gradient magnitude $\kappa$ is large. Second, $\kappa$ should be small to avoid similar overshooting issues. Third, $\epsilon = \|\theta_{\text{ref}} - \theta_*\|$ should be small. This follows from our intuition: a pre-trained reference model $\theta_{\text{ref}}$ serves as a proxy to the optimum $\theta_*$, *prioritizing weight movements toward this proxy can accelerate convergence.*

## B. Grad-Mimic Algorithms

Next, we summarize algorithm steps. Grad-Mimic proceeds in two stages. Stage 1 (Sec. 4.1) computes mimic scores on the fly and updates model weights through re-weighting. Stage 2 (Sec. 4.2) binarizes mimic scores and aggregates utility assessments to obtain the final filtering decisions. Our algorithm supports various training paradigms, including supervised learning and self-supervised learning, as demonstrated in Sec. 5.1 and Sec. 5.2. Moreover, our experiments encompass a wide-range of training configurations, including *full fine-tuning, linear probing, and training models from scratch.*

## C. Experimental Details

We shift the focus from theoretical analysis and algorithm steps to empirical results. We first detail the training configurations and computational resources used in our experiments. We plan to make the source code publicly available upon the publication of this paper.

**Mislabeled Sample Detection.** For the experiments of mislabeled sample detection (Sec. 5.1), we fine-tune ViT-B/16 models pre-trained on the ImageNet-21k dataset (Russakovsky et al., 2015). Training is conducted with a batch size of 32, a learning rate of 1e-4, and the AdamW optimizer. To ensure convergence, we set the training steps to 10 for the DTD and Flowers102 datasets and to 5 for the remaining datasets. We evaluate Grad-Mimic across various temperature values $\tau$ (1.0, 0.9, 0.8, 0.7, 0.6, and 0.5) and report performance at $\tau = 0.5$ in the main paper (Table 1). Results for other temperature values are provided in Table 14. To simulate the reference model, we train ViT-B/16 models on noise-free data split, using different random seeds for weight initialization. All experiments are conducted on an NVIDIA Tesla A100.

---

**Algorithm 1** Online Batch Re-weighting (Stage 1)

---

**Input:** Pre-trained reference model $\boldsymbol{\theta}_{\text{ref}}$, Initial model $\boldsymbol{\theta}_0$, Dataset $\mathcal{D}$, Training steps $T$, Batch size $b$, Learning rate $\eta$, Temperature $\tau$.

**Output:** Normalized mimic scores $\overline{m}_{i,t}$ for all sampled indices $i$

  1: **Initialize:** $\boldsymbol{\theta} \leftarrow \boldsymbol{\theta}_0$
  2: **for** $t = 1$ **to** $T$ **do**
  3:     Sample a mini-batch $\mathcal{B}_t = \{s_i\}_{i=1}^b \sim \mathcal{D}$                                 ▷ Uniform sampling
  4:     $\mathbf{v}_t \leftarrow \boldsymbol{\theta}_{\text{ref}} - \boldsymbol{\theta}_t$                                        ▷ Target direction vector
  5:     **for** each sample $s_i \in \mathcal{B}_t$ **do**
  6:         $\mathbf{g}_{i,t} \leftarrow \nabla_{\boldsymbol{\theta}_t} \ell(s_i; \boldsymbol{\theta}_t)$                       ▷ Compute per-sample gradient
  7:         $m_{i,t} \leftarrow \frac{\langle -\mathbf{g}_{i,t}, \mathbf{v}_t \rangle}{\|\mathbf{v}_t\|}$                    ▷ Project gradient onto target direction
  8:     **end for**
  9:     $\overline{m}_{:,t} \leftarrow \text{Softmax}\left(\frac{m_{:,t}}{\tau}\right)$ where $\overline{m}_{i,t} = \frac{\exp(m_{i,t}/\tau)}{\sum_{j=1}^b \exp(m_{j,t}/\tau)}$     ▷ Softmax normalization over the mini-batch
10:     $\boldsymbol{\theta}_{t+1} \leftarrow \boldsymbol{\theta}_t - \eta \sum_{i=1}^b \overline{m}_{i,t} \mathbf{g}_{i,t}$                    ▷ Re-weighted gradient descent
11: **end for**

---

**Algorithm 2** Offline Sample Selection (Stage 2)

---

**Input:** Normalized mimic score matrix $\overline{\mathbf{M}} = \{\overline{m}_{i,t}\} \in \mathbb{R}^{n \times T}$, Binarization method `mode`, Batch size $b$.

**Output:** Aggregated retain probabilities $p_i$, retained subset $\mathcal{S}$.

  1: Initialize binary decision matrix $\mathbf{B} \leftarrow \mathbf{0}^{n \times T}$
  2: **for** $t = 1$ **to** $T$ **do**
  3:     $\mathbf{m} \leftarrow \overline{\mathbf{M}}_{:,t}$
  4:     **if** `mode` = threshold **then**
  5:         $\tau \leftarrow 1/b$
  6:         $\mathbf{B}_{:,t} \leftarrow \mathbb{I}[\mathbf{m} > \tau]$
  7:     **else if** `mode` = clustering **then**
  8:         Perform 1D $k$-means on $\mathbf{m}$ with $k = 2$
  9:         $\mathbf{B}_{:,t} \leftarrow$ Labels of cluster
10:     **else if** `mode` = top-k **then**
11:         $\tau \leftarrow k$-th percentile of $\mathbf{m}$
12:         $\mathbf{B}_{:,t} \leftarrow \mathbb{I}[\mathbf{m} > \tau]$
13:     **end if**
14: **end for**
15: Train Snorkel `LabelModel` $\mathcal{LM}$ using $\mathbf{B}$ as labeling signals     ▷ Weak Supervision Aggregation
16: $\mathbf{P} \leftarrow \mathcal{LM}.\text{predict\_proba}(\mathbf{B})$                       ▷ Shape: $n \times 2$
17: $p_i \leftarrow \mathbf{P}_{i,1}$ for each sample $i$                 ▷ Extract probability of "Retain"
18: $\mathcal{S} \leftarrow \{i \mid p_i > 0.5\}$                       ▷ Filter subset by decision threshold
19: **return** $\{p_i\}_{i=1}^n, \mathcal{S}$

---

**Data Curation in Large-scale Web Datasets.** For the experiments of multimodal data selection (Sec. 5.2), we follow the training protocol used in DataComp [1] (Cherti et al., 2023; Gadre et al., 2023) and train CLIP models from scratch using contrastive objective on image-caption pairs. Each model is trained for 5 epochs with a batch size of 4096. The total number of seen samples is 12.8M for the small-scale dataset and 128M for the medium-scale dataset.

We analyze different components of the reference model's weights, specifically the final MLP layers in the image and text encoders, respectively. We evaluate Grad-Mimic's effectiveness using temperature values $\tau$ of 0.03, 0.05, 0.07, 0.3, and 0.5. Performance is assessed across 38 diverse downstream tasks (Gadre et al., 2023). Results for $\tau = 0.5$ and $\tau = 0.05$ are reported in the main paper (Table 3), and comprehensive results are provided in Table 13. These experiments are conducted on 8 NVIDIA Tesla A100 GPUs.

---

[1] We identified that some URLs provided by DataComp dataset are now broken. This means that our results might not be comparable to previous approaches on the DataComp Leaderboard. See here for details.

*Table 5.* **Grad-Mimic outperforms other reference model-based methods**: Using reference model last-layer weights is more effective to prioritize samples for learning while reducing substantial computational overheads (see efficiency advantages are in Appendix E).

| | DTD | | | Flowers102 | | | STL10 | | | Oxford-IIIT Pet | | | CIFAR10 | | | CIFAR100 | | | Average | | |
|---|---|---|---|---|---|---|---|---|---|---|---|---|---|---|---|---|---|---|---|---|---|
| Noise Level | 0.4 | 0.5 | 0.6 | 0.4 | 0.5 | 0.6 | 0.4 | 0.5 | 0.6 | 0.4 | 0.5 | 0.6 | 0.4 | 0.5 | 0.6 | 0.4 | 0.5 | 0.6 | 0.4 | 0.5 | 0.6 |
| Mini-Batch SGD | 51.91 | 47.71 | 44.44 | 28.64 | 21.50 | 15.43 | 96.26 | 95.49 | 93.56 | 87.33 | 85.55 | 83.51 | 92.86 | 92.14 | 91.19 | 75.56 | 74.38 | 73.25 | 72.09 | 69.46 | 66.40 |
| Rho-Loss | 54.10 | 52.39 | 48.46 | 41.55 | 35.71 | 30.02 | 97.10 | 96.83 | 96.49 | 88.53 | 87.79 | 86.86 | 94.07 | 93.83 | **93.82** | 76.82 | 76.05 | 75.93 | 75.36 | 73.77 | 71.93 |
| **Grad-Mimic** | **54.68** | **54.10** | **50.43** | **42.75** | **37.10** | **31.71** | **97.16** | **97.00** | **96.90** | **88.80** | **88.25** | **87.14** | **94.15** | **93.92** | 93.80 | **77.24** | **76.82** | **76.05** | **75.80** | **74.53** | **73.01** |

*Table 6.* **Grad-Mimic outperforms Rho-Loss under both setups**: using full or hold-out training set to the build reference model.

| | CIFAR10 | CIFAR100 |
|---|---|---|
| *Using full training set* | | |
| Rho-Loss | 93.83 | 76.05 |
| Grad-Mimic | **93.92** | **76.82** |
| *Using hold-out training set* | | |
| Rho-Loss | 93.26 | 75.11 |
| Grad-Mimic | **93.38** | **75.81** |

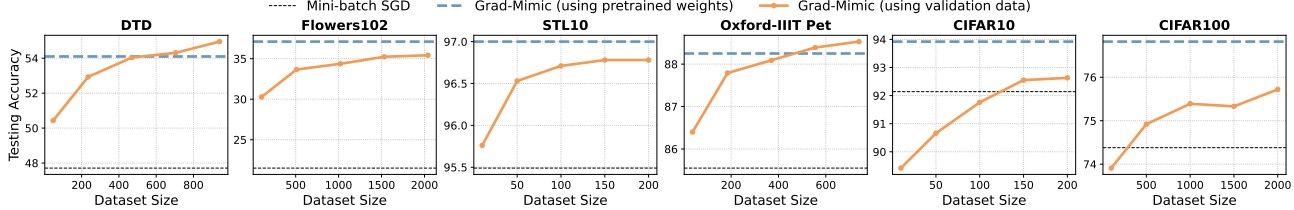

*Figure 7.* **Grad-Mimic outperforms approximated influence function-based methods:** Using reference model's geometry positioning as selection guide offer advantages on their effectiveness and greater accessibility.

# D. Comparing to Model-based Approaches

We provide a further discussion when comparing Grad-Mimic to model-based techniques, specifically reference model-based approaches and influence function-based methods.

**Reference Model-Based Approaches.** We compare Grad-Mimic to Rho-Loss (Mindermann et al., 2022), a representative method that prioritizes samples based on excess loss, computed as the difference between the loss on the current model $\ell(\theta_t)$ and the loss on a pre-trained reference model $\ell(\theta_{\text{ref}})$. For a fair comparison, we simulate pre-trained reference models using two approaches: (i) training on a noise-free hold-out dataset as used in Rho-Loss, and (ii) first training on the entire training dataset. We follow our mislabeled sample detection setup and evaluate the model performance on the testing datasets.

We first demonstrate model performance using reference model that is trained on entire training dataset. As shown in Table 5, Grad-Mimic consistently outperforms Rho-Loss across all datasets and noise levels. Next, we benchmark both methods on CIFAR10 and CIFAR100 with 50% label noise, using a reference model trained on a noise-free hold-out split. The latter setup simulates more realistic scenarios where the reference model, ***acting as a distilled or weaker approximation of ideal weights, is used.*** The results, presented in Table 6, again demonstrate that Grad-Mimic achieves higher testing accuracy than Rho-Loss.

Unlike Rho-Loss, which measures sample utility through differences in *loss space*, Grad-Mimic instead operates in weight space, focusing in particular on last-layer weight discrepancies. This perspective avoids the additional inference passes required by Rho-Loss and instead relies on a simpler and more computationally efficient operation—matrix subtraction.

**Influence Function-based Approaches.** Data influence functions provide a principled way of evaluating the impact of individual training samples on model parameters (Hampel, 1974; Koh & Liang, 2017). However, they require computing the inverse Hessian of the training loss, which is computationally expensive and often limits their practical applicability. Recent studies have explored efficient approximations to estimate sample influence by measuring gradient alignment with a high-quality validation set (Xia et al., 2024; Wu et al., 2024; Wang et al., 2024a). We adapt these methods within Grad-Mimic

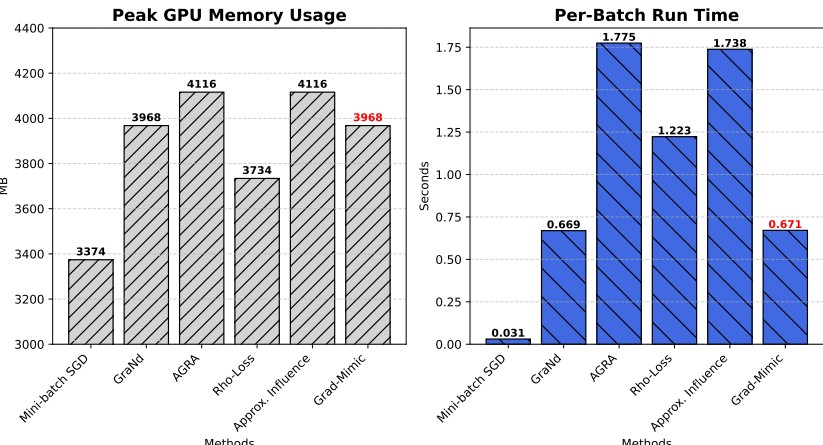

*Figure 8.* Grad-Mimic incurs lower computational cost (reduced runtime) and smaller memory usage than other model-based baselines.

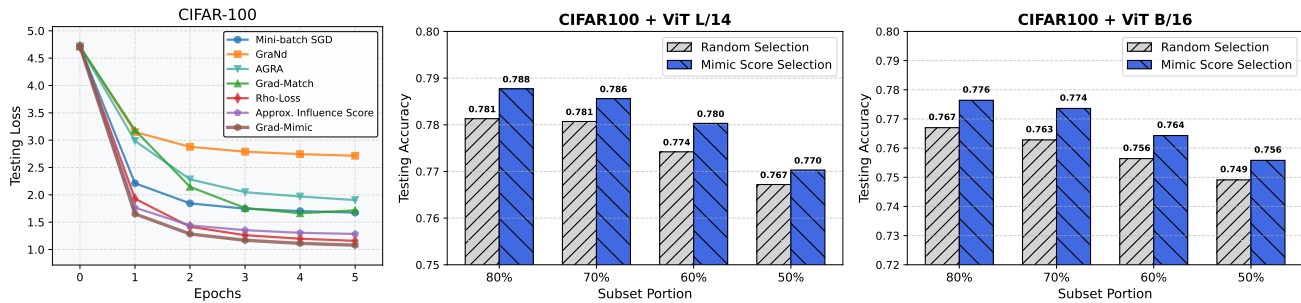

*Figure 9.* Grad-Mimic leads to the fastest convergence and improved data utilization (20% training data can be reduced).

framework by replacing the vector induced by reference model's weights ($v_t$) with *gradients computed from a noise-free validation dataset*. We evaluate these adaptations on six image datasets by adding 50% label noise and assess the impact of validation set size by varying the number of samples drawn from per class (using 1, 5, 10, 15, and 20).

Results in Figure 7 show that larger validation sets improve the accuracy of influence estimates, enabling models to better focus on high-value samples and thereby achieve higher performance. However, Grad-Mimic, which leverages the positioning of pre-trained reference weights, consistently outperforms influence-based approximations. Moreover, relying on labeled validation sets introduces dataset dependencies, incur computational overheads from validation gradient computations, and poses challenges in ensuring their quality, as noted in the limitations of GREATS (Wang et al., 2024a). For example, in the Flowers102 and CIFAR100 datasets, achieving satisfactory performance required over 2,000 correctly labeled samples—***yet still fell short of Grad-Mimic's results***.

## E. Efficiency Advantages

We provide four distinct lines of evidence for the efficiency gains in our technique. They are (i) *computational efficiency*, (ii) *memory usage*, (iii) *learning efficiency*, and (iv) *data utilization*.

**Computational Efficiency and Memory Usage.** First, we benchmark *per-batch training runtime* and *peak GPU memory consumption* across considered baselines, as presented in Figure 8. While mini-batch SGD requires the lowest runtime and memory usage, this involves *no data selection* and leads to suboptimal performance. Grad-Mimic achieves computational runtime and memory usage comparable to GraNd, which prioritizes samples solely based on the norm of training gradients. In contrast, methods like AGRA and approximated influence function-based approaches incur higher computational costs due to additional gradient computations from either another training batch or a noise-free validation dataset. ***This creates a 2.6 times computational overheads than Grad-Mimic***. Similarly, Grad-Mimic outperforms Rho-Loss in efficiency, as Rho-Loss requires performing additional forward passes. Regarding memory usage, Rho-Loss can exceed all these methods

*Table 7.* FLOP comparison of data selection methods per training batch.

| Method | Total FLOPs per batch | Extra overhead vs. SGD |
|---|---|---|
| Mini-batch SGD | $n \cdot (F + B) = 3nF$ | — |
| **Grad-Mimic** | $3nF + 2d_L + 2n \cdot d_L$ | $+(2n + 2) \cdot d_L$ |
| Grad-Norm | $3nF + n \cdot d$ | $+n \cdot d$ |
| AGRA | $2n \cdot (F + B) + 2n \cdot d = 6nF + 2n \cdot d$ | $+3nF + 2n \cdot d$ |
| Rho-Loss | $n \cdot (F + B) + n \cdot F = 4nF$ | $+nF$ |
| Influence function | $(n + m) \cdot 3F + 2nm \cdot d$ | $+3mF + 2nm \cdot d$ |

if the reference model size scales significantly.

Second, we provide a FLOP analysis to formally justify Grad-Mimic's efficiency. Let $F$ denote the FLOPs for one forward pass (using single sample), $B \approx 2F$ for one backward pass, $d$ the full model parameter count, $d_L$ the last-layer parameter count ($d_L \ll d$), $n$ the training batch size, and $m$ the validation set size (used by influence function methods only). Grad-Mimic's extra cost over standard SGD consists of three terms, all operating over $d_L$ parameters only: (1) one weight subtraction $v_t = \theta_{\text{ref}} - \theta_t$ ($d_L$ FLOPs, once per batch), (2) one $\ell_2$ norm $\|v_t\|$ ($d_L$ FLOPs, once per batch), and (3) one projection $\langle -g_i, v_t \rangle / \|v_t\|$ per sample ($2d_L$ FLOPs each for the dot product; the scalar division is negligible). The total extra cost is $(2n + 2) \cdot d_L$.

Results are shown in Table 7. Grad-Mimic compares to baselines:

- **vs. Grad-Norm**: GraNd computes $\|g_i\|$ over all $d$ parameters per sample, adding $n \cdot d$ FLOPs.
- **vs. AGRA**: AGRA draws a second batch of $n$ samples and runs full backward passes, then computes cosine similarity over all $d$ parameters, adding $3nF + 2n \cdot d$ FLOPs.
- **vs. Rho-Loss**: Rho-Loss performs an additional forward pass through the reference model for every training sample ($+nF$).
- **vs. Influence functions**: The dominant term $2nm \cdot d$ grows multiplicatively with batch size, validation set size, and model width simultaneously. As $m$ increases for better influence estimates, cost compounds.

Our efficiency stems from the design choice to measure sample utility using only the last-layer weight difference, which we find sufficient for effective data selection. For example, in large-scale CLIP pretraining, the corresponding weight matrix has dimensions $3072 \times 768$. Computing the weight subtraction becomes a lightweight operation compared to standard training steps.

**Learning Efficiency.** Next, we evaluate learning efficiency by analyzing convergence curves. As shown in Figure 9 (left), experiments on CIFAR100 with 50% label noise demonstrate that Grad-Mimic converges faster than all the compared methods, achieving higher model performance within a fixed number of training epochs. Additionally, Figure 5 (left) illustrates that, on the DataComp medium-scale dataset (over 100 million samples), Grad-Mimic converges faster than standard training, *reducing training steps by 20.7%*.

**Data Utilization.** We introduce *a new experiment* that measures data utilization without adding label noise. Instead of simulating mislabeled samples, we rank *a clean training dataset* by their mimic scores and build subsets from the top-k% entries. We use a well-trained ViT-B/16 as our reference and perform Grad-Mimic on this clean CIFAR100 dataset. We evaluate the resulting model performance trained on refined subsets. Results, presented in Figure 9, show that Grad-Mimic's selected subsets consistently outperform randomly drawn subsets of the same size. Remarkably, with both ViT-B/16 and the larger ViT-L/14, Grad-Mimic *matches the full-dataset accuracy while reducing the training data by 20%. This highlights a direct reduction in data-related computational cost.*

We note that this experiment assumes access to a fully clean dataset—*an assumption that is often unrealistic in practice*. Our main paper focuses on more challenging and practical settings involving mislabeled samples, where effective tools for identifying low-value samples are typically unavailable.

These findings—*lower runtime per batch, reduced GPU memory usage, faster convergence, and improved data utilization*—demonstrate that Grad-Mimic achieves significant efficiency gains, supporting our motivation.

*Table 8.* **Mislabeled Sample Detection with Alternative Aggregation Methods**: F1-scores demonstrate identification capability across six datasets. Weak supervision techniques outperforms naive aggregation method—majority vote.

|  | CIFAR100 | CIFAR10 | DTD | Flowers102 | Oxford-IIIT Pet | STL10 |
|---|---|---|---|---|---|---|
| Majority Vote | 90.2 | 90.1 | 89.4 | 93.5 | 90.5 | 91.3 |
| Snorkel | 96.2 | 94.0 | **97.0** | 97.1 | 97.5 | 95.3 |
| FlyingSquid | 96.7 | 94.0 | **97.0** | **97.6** | 97.5 | 97.3 |
| UWS | **97.4** | **97.8** | 92.6 | **97.6** | **99.0** | **98.9** |

*Table 9.* **Runtime Comparison of Stage 2 Aggregation Methods**: Results are measured in seconds. UWS achieves competitive performance with significantly lower computational cost.

|  | CIFAR100 (50K) | CIFAR10 (50K) | DTD (1.9K) | Flowers102 (1K) | Oxford-IIIT Pet (3.7K) | STL10 (5K) |
|---|---|---|---|---|---|---|
| Majority Vote | 0.68 | 0.66 | 0.07 | 0.05 | 0.09 | 0.11 |
| Snorkel | 0.84 | **0.32** | 0.13 | 0.13 | 0.14 | 0.14 |
| FlyingSquid | 0.76 | 0.76 | 0.06 | 0.03 | 0.06 | 0.08 |
| UWS | **0.28** | 0.32 | **0.02** | **0.01** | **0.02** | **0.03** |

## F. Aggregation Strategy Analysis

Next, we present an additional study evaluating well-known aggregation models commonly used in weak supervision. Specifically, we examine naive aggregation through majority vote, Snorkel (Ratner et al., 2017), FlyingSquid (Fu et al., 2020), and UWS (Shin et al., 2021).

**Comparison of Aggregation Methods.** We follow the same experimental protocol as in Table 2 to assess the performance of these methods used in Grad-Mimic Stage 2. Label noise is injected into 50% of the samples in each dataset, and GMM clustering is then applied as a representative binarization method to convert estimated sample utilities into binary retain/discard votes.

Results, shown in Table 8, indicate that weak supervision-based methods (Snorkel, FlyingSquid, UWS) consistently outperform majority vote. This supports our motivation to ***denoise signals from inaccurate local gradients and capturing the evolving utility of samples throughout training***. Notably, Grad-Mimic combined with any of these weak supervision aggregators consistently identifies incorrect samples across all datasets, achieving F1-scores above 95%.

**Runtime Analysis.** We expect Stage 2 incurs minimal computational overheads compared to the Stage 1 phase. To verify this, we measure the runtime required to aggregate mimic scores and infer filtering decisions for each dataset under different aggregation methods. Results are reported in Table 9. As shown, runtime scales with training dataset size. UWS method achieves the lowest runtime in most cases, while remaining *below one second* across all settings. This supports that Stage 2 offline selection process ***is computationally negligible and introduces no significant overheads.***

**Effect of Training Data Size.** Although runtime scales with the size of the training dataset, the amount of data required to learn an effective aggregator can be substantially reduced. We therefore examine how much data is needed to train an accurate weak supervision label model. Here, we use Snorkel model as example. We randomly select 0.5%, 1%, 5%, and 10% of the samples (along with their stored mimic scores) to train the label model. For each sampling ratio, we conduct five independent trials and report the average F1-scores. Results are displayed in Table 10, which indicate that loading and modeling the full dataset is unnecessary: ***in most cases, using as little as 0.5% of the data achieves performance competitive to training on the entire dataset.*** In other words, we can use limited sample assessments to build a reliable ensemble filter to automate effective data curation.

## G. Predicting Pre-training Dataset.

We explore an interesting application of mimic score: *predicting the pre-training dataset.* Specifically, we ask: *how accurately can Grad-Mimic check whether a given sample was used to train a reference model?* We test this hypothesis on the small-scale DataComp dataset. We first apply various pre-built filters to curate datasets and train CLIP models on each.

*Table 10.* **Effect of Training Data Size in Stage 2 Aggregation**: We show that using only 0.5% of training data achieves competitive performance across datasets.

| Ratio (%) | CIFAR100 (50K) | CIFAR10 (50K) | DTD (1.9K) | Flowers102 (1K) | Oxford-IIIT Pet (3.7K) | STL10 (5K) |
|---|---|---|---|---|---|---|
| **0.5%** | **97.2** | **94.5** | **98.2** | 86.7 | 98.0 | **97.4** |
| 1% | 96.5 | 94.1 | 96.6 | 88.9 | **98.1** | 96.0 |
| 5% | 96.6 | 93.9 | 97.9 | 96.8 | 97.3 | 96.5 |
| 10% | 96.2 | 93.8 | 97.3 | **97.6** | 97.3 | 96.0 |
| 100% | 96.2 | 94.0 | 97.0 | 97.4 | 97.5 | 95.3 |

*Table 11.* **Mimic Score for Data Membership Prediction**: Mimic score-based selection significantly outperforms random selection in identifying samples used to train the reference model.

| Filtering Strategy | Random Selection | | Mimic Score Selection | |
|---|---|---|---|---|
| | Jaccard Similarity | Overlap Percentage | Jaccard Similarity | Overlap Percentage |
| CLIP-ViT B/32 Top 30% | 0.149 | 0.258 | **0.232** | **0.376** |
| CLIP-ViT L/14 Top 30% | 0.150 | 0.260 | **0.236** | **0.382** |
| DataComp'23 Top-ranked | 0.166 | 0.284 | **0.271** | **0.426** |

These creates several reference models to follow. Then, we target their final-layer weights to mimic and apply Grad-Mimic to train *on the entire data pool*. Our goal is to see whether the top-ranked samples (matched in size to the curated datasets) identified by mimic score appear in their training dataset.

We evaluate our approach against random selection using Jaccard similarity and percentage of overlap. As shown in Table 11, mimic score-based selection identifies more samples used to train the reference model compared to random sampling. We achieve a **42.6%** overlap with DataComp'23 best-performing filtered dataset (Huang et al., 2024b) ***by simply mimicking final-layer weights without direct access to their filtering steps.***

## H. Ablation Studies

We present extensive ablation studies of Grad-Mimic. In particular, we analyze the framework robustness under different design choices and the effects of the reference model to address reference model dependency.

**Full Fine-tuning Results.** We report the complete Stage 1 results for the mislabeled sample experiments in Table 12. Beyond the linear probing setting (shown in the main paper), we also consider a full fine-tuning regime where Grad-Mimic targets only the final layer for mimicry while updating all model parameters based on derived mimic scores. In this setup, Grad-Mimic can also effectively guide weight updates across the entire network and ultimately outperforms all baseline methods in average.

**Choice of Temperature.** Next, we study Grad-Mimic under different temperature values and compare them to baseline methods (Mini-batch SGD and Grad-Match). The results for mislabeled sample experiments are presented in Table 14. In these datasets, we set the noise level to 0.3 and fine-tune ViT-B/16 model under linear probing configuration. We observe that lower temperature values generally lead to higher testing accuracy, as they sharpen the normalization of mimic scores and encourage the model to focus more on high-value samples during training. Furthermore, Grad-Mimic consistently outperforms both baseline methods across all temperature settings, demonstrating robustness to temperature choice.

**Extension to LLM and Text-Domain Tasks.** Grad-Mimic is easily extendable to other modalities. We conducted a new set of experiments using Pythia-160M (Biderman et al., 2023) on four text datasets: ARC-Easy (Clark et al., 2018), ARC-Challenge (Clark et al., 2018), SciQ (Johannes Welbl, 2017), and OpenBookQA (Mihaylov et al., 2018). We re-use the setup in Sec. 5.1, we inject noise by flipping ground-truth answers to random distractor options. We use a noise-free validation set (10% of the training set) and create the reference models. We compare Grad-Mimic against standard fine-tuning (SFT) and approximated influence-function-based methods (IF), measuring negative log-likelihood (NLL). We use the fine-tuned checkpoint's performance to show each method's batch reweighting effectiveness. Grad-Mimic yields faster

*Table 12.* **Stage 1 Full Results in Mislabeled Sample Experiment**: Both training configurations validate the claim that using mimic scores can effectively down-weight mislabeled samples during training, improving data efficiency and denoising capability.

| *Full Fine-tuning* | DTD | | | Flowers102 | | | STL10 | | | Oxford-IIIT Pet | | | CIFAR10 | | | CIFAR100 | | | Average | | |
|---|---|---|---|---|---|---|---|---|---|---|---|---|---|---|---|---|---|---|---|---|---|
| Noise Level | 0.4 | 0.5 | 0.6 | 0.4 | 0.5 | 0.6 | 0.4 | 0.5 | 0.6 | 0.4 | 0.5 | 0.6 | 0.4 | 0.5 | 0.6 | 0.4 | 0.5 | 0.6 | 0.4 | 0.5 | 0.6 |
| Mini-batch SGD | 46.91 | 42.02 | 33.24 | 65.54 | 55.90 | 39.84 | 75.35 | 58.57 | 49.98 | 70.43 | 64.62 | 55.63 | 90.31 | 86.69 | 82.01 | 69.42 | 63.77 | **59.28** | 69.66 | 61.93 | 53.33 |
| GraNd | 29.73 | 8.83 | 15.43 | 10.80 | 2.78 | 27.92 | 78.39 | 52.32 | 64.30 | 56.83 | 45.38 | 45.43 | 87.56 | 83.79 | 80.04 | 18.65 | 8.83 | 7.84 | 46.99 | 33.66 | 40.16 |
| AGRA | 46.28 | 37.29 | 32.39 | 0.62 | 0.93 | 0.82 | **84.35** | 79.94 | 78.17 | 76.97 | 77.11 | 69.39 | 89.23 | 86.79 | 68.09 | 11.96 | 8.44 | 6.88 | 51.57 | 48.42 | 42.62 |
| Grad-Match | 48.88 | 42.66 | 33.24 | 37.92 | 54.94 | 42.62 | 83.86 | **81.16** | 77.18 | 74.35 | 61.73 | 51.16 | 44.48 | 35.69 | 37.59 | 69.41 | 66.21 | 20.09 | 58.82 | 57.07 | 43.65 |
| **Grad-Mimic** | 49.20 | 42.82 | 33.83 | 68.58 | 56.46 | 44.27 | 72.06 | 71.85 | 83.09 | 81.98 | 78.30 | 73.34 | 90.52 | 89.07 | 66.41 | 73.97 | 74.31 | 24.02 | 72.72 | 68.60 | 54.16 |

| *Linear Probing* | DTD | | | Flowers102 | | | STL10 | | | Oxford-IIIT Pet | | | CIFAR10 | | | CIFAR100 | | | Average | | |
|---|---|---|---|---|---|---|---|---|---|---|---|---|---|---|---|---|---|---|---|---|---|
| Noise Level | 0.4 | 0.5 | 0.6 | 0.4 | 0.5 | 0.6 | 0.4 | 0.5 | 0.6 | 0.4 | 0.5 | 0.6 | 0.4 | 0.5 | 0.6 | 0.4 | 0.5 | 0.6 | 0.4 | 0.5 | 0.6 |
| Mini-batch SGD | 51.91 | 47.71 | 44.44 | 28.64 | 21.50 | 15.43 | 96.26 | 95.49 | 93.56 | 87.33 | 85.55 | 83.51 | 92.86 | 92.14 | 91.19 | 75.56 | 74.38 | 73.25 | 72.09 | 69.46 | 66.40 |
| GraNd | 36.22 | 30.59 | 25.32 | 18.23 | 13.90 | 10.49 | 68.70 | 59.55 | 49.68 | 69.47 | 60.02 | 48.13 | 69.47 | 60.02 | 48.13 | 71.60 | 70.51 | 69.27 | 58.71 | 53.36 | 47.40 |
| AGRA | 41.81 | 36.28 | 31.49 | 41.81 | 36.28 | 31.49 | 96.19 | 95.15 | 93.11 | 85.25 | 82.53 | 78.39 | 92.51 | 92.01 | 90.99 | 72.50 | 71.30 | 69.69 | 71.68 | 68.93 | 65.86 |
| Grad-Match | 51.81 | 47.55 | 41.33 | 27.00 | 20.21 | 14.90 | 96.06 | 95.34 | 93.44 | 86.86 | 85.75 | 83.18 | 92.62 | 92.04 | 91.17 | 75.39 | 74.46 | 73.18 | 71.62 | 69.23 | 66.20 |
| **Grad-Mimic** | 54.68 | 54.10 | 50.43 | 42.75 | 37.10 | 31.71 | 97.16 | 97.00 | 96.90 | 88.80 | 88.25 | 87.14 | 94.15 | 93.92 | 93.80 | 77.24 | 76.82 | 76.05 | 75.80 | 74.53 | 73.01 |

*Table 13.* **Full Stage 1 Results in DataComp Experiment**: On both dataset scales, with the aid of publicly available pre-trained weights, Grad-Mimic consistently improves CLIP model performance across all temperature settings.

| Scale | Training Method | Mimic Layer $\theta_{\text{ref}}$ | Temperature $\tau$ | ImageNet | ImageNet dist. shifts | VTAB | Retrieval | Average over 38 datasets (↑) |
|---|---|---|---|---|---|---|---|---|
| Small | Vanilla Training | — | — | 0.026 | 0.035 | 0.139 | 0.114 | 0.131 |
| | Grad-Mimic | Last MLP Layer in Text Encoder | 0.03 | 0.026 | 0.035 | 0.153 | 0.115 | **0.136** |
| | | | 0.05 | 0.026 | 0.035 | 0.147 | 0.114 | **0.135** |
| | | | 0.07 | 0.026 | 0.035 | 0.151 | 0.116 | **0.135** |
| | | | 0.3 | 0.026 | 0.034 | 0.154 | 0.112 | **0.137** |
| | | | 0.5 | 0.027 | 0.035 | 0.152 | 0.114 | **0.139** |
| | | Last MLP Layer in Image Encoder | 0.03 | 0.025 | 0.035 | 0.149 | 0.114 | **0.133** |
| | | | 0.05 | 0.026 | 0.037 | 0.162 | 0.118 | **0.146** |
| | | | 0.07 | 0.027 | 0.036 | 0.166 | 0.118 | **0.145** |
| | | | 0.3 | 0.026 | 0.035 | 0.161 | 0.114 | **0.144** |
| | | | 0.5 | 0.027 | 0.035 | 0.160 | 0.117 | **0.145** |
| Medium | Vanilla Training | — | — | 0.171 | 0.148 | 0.253 | 0.217 | 0.254 |
| | Grad-Mimic | Last MLP Layer in Image Encoder | 0.05 | 0.169 | 0.151 | 0.262 | 0.216 | **0.258** |

*Table 14.* **Effect of Temperature in Stage 1 Simulation Experiment**: Grad-Mimic consistently outperforms baselines across all temperature values, with lower temperatures yielding better testing accuracy by focusing on high-value samples

| Method | DTD | Flowers102 | Oxford-IIIT Pet | CIFAR10 | CIFAR100 |
|---|---|---|---|---|---|
| SGD | 52.8 | 35.6 | 87.9 | 93.2 | 76.2 |
| Grad-Match | 53.1 | 32.1 | 87.4 | 93.0 | 76.1 |
| *Grad-Mimic with varying temperature $\tau$* | | | | | |
| $\tau = 1.0$ | **56.5** | 44.9 | 89.0 | 94.1 | 77.0 |
| $\tau = 0.9$ | 56.4 | 45.3 | 89.1 | 94.1 | 77.1 |
| $\tau = 0.8$ | 56.3 | 45.8 | 89.1 | 94.1 | 77.2 |
| $\tau = 0.7$ | 56.3 | 46.1 | 89.2 | **94.2** | 77.2 |
| $\tau = 0.6$ | 56.0 | **46.4** | 89.3 | **94.2** | **77.3** |

convergence than SFT and lower NLL than competing methods. This confirms that Grad-Mimic is architecture-agnostic by design: Mimic Score depends only on gradient projections and the last-layer weight differences, neither of which is modality-specific.

**Which layer should be mimicked?** We study the impact of the layer weights to mimic. We vary the depth of the reference

*Table 15.* Batch reweighting on text-Domain tasks, measured by NLL ($\downarrow$).

| | 20% Label Noise | | | 30% Label Noise | | |
|---|---|---|---|---|---|---|
| Dataset | SFT | Influence Function | Grad-Mimic | SFT | Influence Function | Grad-Mimic |
| **ARC Challenge** | 3.463 | 3.458 | **3.440** | 3.471 | 3.458 | **3.445** |
| **ARC Easy** | 3.207 | 3.271 | **3.181** | 3.207 | 3.271 | **3.196** |
| **OpenbookQA** | 5.209 | 6.100 | **5.090** | 5.240 | 6.100 | **5.104** |
| **SciQ** | 1.528 | 1.621 | **1.456** | 1.607 | 1.659 | **1.561** |

*Table 16.* Last-layer weights consistently yield the best performance across all datasets.

| | DTD | Oxford-IIIT Pet | CIFAR10 | CIFAR100 |
|---|---|---|---|---|
| Layer 8 | 57.5 | 75.5 | 90.7 | 75.2 |
| Layer 9 | 58.3 | 75.5 | 90.8 | 75.3 |
| Layer 10 | 58.6 | 74.0 | 91.1 | 74.9 |
| Layer 11 | 58.2 | 75.1 | 90.2 | 75.3 |
| **Layer 12 (last layer)** | **59.5** | **75.8** | **91.2** | **75.5** |

model weights used to obtain the target vector and evaluate the resulting model performance. Results are presented in the Table 16. Grad-Mimic's performance is robust across a wide range of deep layers (Layers 8 $\sim$ 12) with minimal performance variance. Additionally, we find that mimicking the last-layer weights yields the highest accuracy in most cases. While the optimal choice may depend on architecture or task, these findings suggest that later representations generally offer the richer information and are better choices for estimating sample utility.

**Generalization to Different Reference Models.** We examine whether Grad-Mimic overfits to a specific reference model by evaluating its sensitivity to reference model variation. We initialize and train multiple reference models with different random seeds, producing high-performing models in distinct regions of the weight space. Our online re-weighting algorithm is then used to train a separate model (also initialized with a different seed) on a dataset with 50% mislabeled samples, which targets different created reference weights. We evaluate the resulting model with their corresponding reference and compute the performance variance. As shown in Table 17, the variance is *extremely low*, indicating that Grad-Mimic generalizes well across reference models. In other words, this addresses concerns about *reference model dependency, as performance remains consistent regardless of which high-performing reference model is used*.

**Impact of Reference Model Quality.** Next, we investigate how the quality of the reference model impacts the effectiveness of Grad-Mimic, particularly in scenarios where *only a weaker or distilled model is available*. To simulate such scenarios (e.g., using early checkpoints, working on mismatched tasks, or following resource-constrained training recipe), we degrade the reference model in two controlled ways:

- **Weight corruption**: we inject zero-mean Gaussian noise $(0, \sigma^2)$ into the reference model's weights. By varying the variance of the added noise, we analyze the performance of models trained with Grad-Mimic against standard training methods that do not leverage a reference model.
- **Data-limited training**: we train the reference model on smaller fractions of its original dataset (shrinking from 1.3B samples to 3.5M samples). Fewer training samples will produce a weaker (or distilled) reference model.

We assess Grad-Mimic in both the mislabeled sample and multimodal settings, applying each degradation method separately. We present these two analysis in Figure 10 and Table 18.

As expected, increasing noise variance or reducing the amount of training data degrades the performance of the reference model, which in turn lowers the effectiveness of Grad-Mimic. Nevertheless, even under significant noise (heavy degradation), models trained with Grad-Mimic still outperform Mini-batch SGD and, in some cases, *even exceed the performance of the degraded reference model itself* (Figure 10). These results suggest that Grad-Mimic is not merely performing distillation. Instead, it leverages the reference model's geometric positioning to define a generally beneficial optimization direction, enabling the trained model to surpass the reference despite its imperfections.

Moreover, even when the reference model is trained on *372$\times$ less data (3.5M vs. 1.3B samples)*, Grad-Mimic remains highly

*Table 17.* **Generalization to Different Reference Models**: Grad-Mimic achieves consistent performance across different reference models, demonstrating low variance and strong robustness.

| Reference Model Seed | CIFAR100 | CIFAR10 | STL10 | Oxford-IIIT Pet | DTD |
|---|---|---|---|---|---|
| 123 | 75.60 | 93.67 | 96.68 | 87.84 | 53.51 |
| 213 | 75.64 | 93.69 | 96.70 | 86.37 | 50.69 |
| 231 | 76.01 | 93.66 | 96.65 | 86.45 | 51.01 |
| 312 | 75.59 | 93.79 | 96.68 | 86.37 | 50.69 |
| 321 | 75.63 | 93.63 | 96.95 | 86.40 | 50.69 |
| **Variance** | **2.53e-06** | **2.98e-07** | **1.24e-06** | **3.35e-05** | **1.21e-04** |

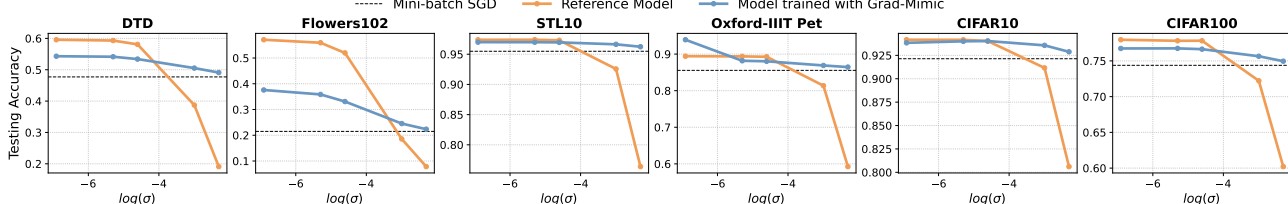

*Figure 10.* Grad-Mimic's performance is robust to the quality of reference model.

*Table 18.* **Performance with Weaker Reference Models**: Grad-Mimic achieves competitive performance even when the reference model is trained on significantly fewer samples.

| Training Dataset | Method | ImageNet | ImageNet dist. shifts | VTAB | Retrieval | Average over 38 datasets (↑) |
|---|---|---|---|---|---|---|
| 1.3B samples | Vanilla Training | 0.026 | 0.035 | 0.139 | 0.114 | 0.131 |
| 1.3B samples | Grad-Mimic | 0.026 | 0.037 | 0.162 | 0.118 | 0.146 |
| **3.5M samples (372× fewer)** | **Grad-Mimic** | **0.026** | **0.035** | **0.158** | **0.115** | **0.140** |

competitive. As shown in Table 18, Grad-Mimic with this significantly smaller proxy achieves an average performance of 0.140, which still outperforms the 0.131 average of vanilla training. These results address concerns about strong reference model dependency and highlight that *while a high-performing reference model is ideal, a weaker proxy can still provide valuable guidance and improve training over traditional baselines.*

**New Domain Reference Models.** A natural question arises when applying Grad-Mimic to a new or highly specialized domain where no high-performing reference model is readily available. This cold-start limitation is common in reference model-based methods (Lin et al., 2024b; Mindermann et al., 2022). While such scenarios are inevitable, they can be easily mitigated by first training a reference model on a relatively small, hold-out dataset.

Our experiments on mislabeled sample detection validate this approach. In particular, for out-of-domain tasks such as DTD (texture recognition) and Flowers102 (fine-grained flower classification), we first create reference models by fine-tuning on their noise-free hold-out datasets. Despite the domain gap, as shown in Table 1, Grad-Mimic consistently outperforms competing methods on average.

**Compared to Human-designed Filters.** Finally, we compare mimic score-based filters against *basic filtering strategy* in our DataComp experiment. This competing filter removes samples based on human-defined criteria such as caption length, image size, and caption language, representing the human-designed approach, which requires expensive trial-and-error to establish. We use the medium-scale dataset as our testbed and present the results in Table 19. As shown, while basic filtering improves performance from 0.254 to 0.267 by reducing the dataset from 101.9M to 30M samples, our mimic score-based filter (top 30% selection) achieves further improvement to 0.268 using 1M fewer samples (29M total).

*Table 19.* **Comparison with Human-Designed Filters**: Mimic score-based filtering outperforms basic filtering (human-designed heuristics) while using fewer samples (one million samples less).

| Filtering Strategy | Dataset Size | Average over 38 datasets (↑) |
|---|---|---|
| No Filtering | 101.9M | 0.254 |
| Basic Filtering | 30M | 0.267 |
| **Mimic Score Top 30%** | **29M (1M ↓)** | **0.268** |

## I. Discussion and Future Work.

**Reference Model Dependency.** In this work, we argue that the reference model dependency ***opens up new opportunities that competing methods cannot exploit***. **First**, the modern AI landscape is characterized by a strong *access asymmetry*: well-trained models are routinely released publicly, while the carefully curated datasets used to produce them remain proprietary. This trend works directly in our favor—the more powerful public models become, the more powerful Grad-Mimic's guidance signal becomes. **Second**, even in the cold-start scenario, a reliable proxy can be obtained by training on a smaller hold-out split until convergence—precisely the setup in our mislabeled sample detection experiments. Table 18 validates that a reference trained on $372\times$ fewer web-crawled samples (3.5M vs. 1.3B) still allows Grad-Mimic to outperform standard training. Our experiments on LLM and text-domain datasets also confirm this (see Table 15): a reference model trained on a 10%-sized validation set can guide training that consistently outperforms standard fine-tuning and influence-function-based methods. These all confirm that the bar for a useful reference is surprisingly *low*. **Third**, Grad-Mimic is *robust* to reference model quality. Injecting increasing levels of Gaussian noise into the reference weights still sees Grad-Mimic outperforms Mini-batch SGD (see Figure 10), suggesting it *exploits geometric positioning rather than requiring a perfect reference*. **Finally**, *model dependency is strictly easier to satisfy than data dependency*—and unlocks strictly more. Influence function-based methods require a carefully curated validation set with additional gradient computations, expensive in both data quality and compute. Grad-Mimic simply requires a publicly available checkpoint and the last-layer weight subtraction, transforming a freely available resource into actionable data-quality insights.

We outline several promising directions that extend the scope and impact of this work, including *cross-architecture alignment, data marketplace design, and membership inference*.

**Cross-Architecture Alignment.** Throughout this work, we assume the reference model and the model being trained *share identical architectures*, enabling direct weight-space comparison through simple subtraction. However, broader scenarios involve leveraging reference models with different architectures. For example, practitioners may wish to use a large model (e.g., ViT-L/14) to guide the training of a smaller, more efficient variant (e.g., ViT-B/16). Addressing this setting requires principled alignment of weight spaces across architectures. Several potential approaches exist, including learning projection mappings (Roberts et al., 2024), weight interpolation (Ainsworth et al., 2022), and stitching model blocks (Stoica et al., 2023). Our framework is compatible with these weight alignment techniques, which permit data-quality distillation from diverse model architectures.

**Data Marketplace Design.** Mimic score provides an efficient measuring tool for individual sample utility to train a new model. This naturally connects to the field of data marketplaces, where organizations buy and sell training data and a principled valuation mechanism is essential for fair pricing, quality assurance, and incentive design (Huang et al., 2023b; Chen et al., 2019; Jagadeesan et al., 2023; Chen et al., 2022; Fernandez et al., 2020; Azcoitia & Laoutaris, 2022). Our approach raises several important research questions: How stable are mimic scores across different reference models, particularly when buyers target diverse downstream domains simultaneously? How does the quality or capacity of the reference model affect valuation reliability?

Our preliminary results in Appendix H provide promising evidence, demonstrating strong generalization across reference models and robustness even under strongly degraded references. Building on these, future work could explore market mechanisms where mimic scores are used to derive bid and ask prices for data, enabling principled price negotiation between buyers and sellers.

**Membership Inference.** Our preliminary exploration in Appendix G suggests that mimic scores can partially predict which individual samples were used to train a reference model, outperforming random selection and achieving 42.6% overlap with known training sets. This observation connects Grad-Mimic to the literature on membership inference, which aims to

determine whether a specific sample was part of a model's training set—an issue with significant implications for privacy, data provenance, and train-test contamination (Shi et al., 2023; Golchin & Surdeanu, 2023; Oren et al., 2023).

Grad-Mimic departs from prior membership inference methods that rely on overfitting signals such as abnormal confidence scores. Instead, mimic scores exploit weight-space geometry, measuring a sample's directional compatibility through its gradients and a vector pointing to the reference model. This geometric perspective offers a fundamentally different mechanism for revealing training set membership and may complement existing inference techniques.

