# OpenReview forum: "Evaluating Sample Utility for Efficient Data Selection by Mimicking Model Weights"
_ICML.cc/2026/Conference — ICML 2026 regular_

### Official Review · Reviewer_8TiY · 2026-03-05

**Soundness:** 2
**Presentation:** 3
**Significance:** 2
**Originality:** 2
**Overall Recommendation:** 3
**Confidence:** 3

**Summary:**

This paper mainly solves the problem of how to quickly select the most useful data for model training from massive, noisy web data. Previous methods either relied on manually written rules (too slow and inflexible) or calculated how much each data point affects the model (too expensive and computationally heavy). The authors designed a quick method called Mimic Score. First, the authors took a well-trained, high-performance model as the reference model. Assume that this reference model's parameter direction is the desired direction. When training a new model, the authors compute the score between a sample's negative gradient and the direction (\theta_ref - \theta). These scores are utilized to reweight or select the training samples. Extensive experiments verify the good performance of the mimic score-based weighting and selection.

**Compliance With Llm Reviewing Policy:**

Affirmed.

**Final Justification:**

I still maintain that the novelty of this paper is seriously insufficient (which caused me to overlook the theoretical analysis in the appendix initially), and I remain skeptical about the effectiveness of the reference model, the lack of more solid theoretical support compared to validation‑gradient‑based methods, and the lack of diversity (merely adding one experiment is not convincing enough). However, given that other reviewers have given positive assessments, I will lower my confidence and let the Area Chair make the final decision.

**Key Questions For Authors:**

1. There are many studies which also apply the gradients for data valuation. Please compare the differences between yours and other related studies.
2. The data diversity is ignored in this study. Does it have little influence on data reweighting or selection?

**Limitations:**

A good data selection strategy should also take diversity into account. Therefore, future work could explore a more balanced framework that considers both utility and diversity to further enhance the reliability of data selection.

**Strengths And Weaknesses:**

In my opinion, the strengths of this work are:
1） It is very simple and intuitive.
2） It supports both weighting and selection, making it flexible in practice.
However, it also has significant drawbacks:
1） It lacks theoretical justification for why is this method valid? In fact, gradient-based data valuation has been widely studied, where the score is typically defined as the inner product between the sample gradient and the validation gradient.This paper essentially uses the difference between a proxy model and the current model to approximate the validation gradient.Although this is intuitively reasonable, there is no theoretical analysis. Furthermore, during training, the ideal validation gradient changes dynamically with the training process. Using a fixed model difference as its approximation is not fully justified.
2) Most existing work on data weighting or selection considers not only utility but also diversity. This paper does not mention diversity at all, which is a major flaw.

---

> ### Author Rebuttal · Authors · 2026-03-30
>
> We thank the reviewer for recognizing the **simplicity, intuition, and flexibility of our framework**. Below, we provide clarifications regarding our theoretical justification and related work, followed by a discussion on data diversity.
>
> ---
>
> * **On Theoretical and Technical Justification**:
>     * We have provided **a formal convergence analysis in Appendix A**. We proved that by leveraging weight-space geometry for batch re-weighting, our strategy can achieve faster convergence than stochastic gradient descent when handling noisy datasets.
>     * **Weight Difference vs. Validation Gradient**: We clarify that our method **is not intended to approximate the validation gradient**. Relying on validation gradient **introduces dependencies on dataset quality and significant computational overheads**. These limitations happen in prior influence-function literature. Our approach sidesteps these by using a dataset-agnostic weight-space difference. Moreover, empirically, **Figure 3 shows that Grad-Mimic outperforms influence-function-based methods. Our new rebuttal results in LLM setups also confirm this**. Theoretically, we provided another analysis bounding the difference between the target vector and validation gradient to justify its robustness. This study is provided [here](https://anonymous.4open.science/r/Mimic_Rebuttal-0FE1).
>     * Finally, we clarify that **we are not using a fixed weight difference throughout training**. The target vector is computed by $v_t := \theta_{ref} - \theta_t$, which is a **time-varying vector** that dynamically adjusts based on the model’s current state.
>
> * **On Related Work Discussion**:
>     * We have compared a broad range of gradient-based valuation methods (including Grad-Norm, AGRA, Grad-Match, and Influence-function-based methods). Our empirical results in Table 1 and Figure 3 highlight the following key distinctions:
>         * **vs. Influence Function Methods**: These methods approximate sample influence via a noise-free validation set, **introducing dataset dependencies, additional gradient computations, and data curation challenges**. Grad-Mimic replaces this with a weight-space vector derived from a publicly available model. As shown in Figure 3, on Flowers102 and CIFAR100, influence-based methods required over 2,000 correctly labeled samples yet still fell short of Grad-Mimic's performance.
>         * **vs. AGRA / Grad-Match**: These rely on gradient similarities across batches, making them sensitive to noisy gradients and costly due to inner-batch comparisons. Grad-Mimic instead uses weight-space geometry as a proxy oracle signal, delivering higher performance (Table 1) and smaller compute overhead (Figure 8).
>         * **vs. Grad-Norm**: Grad-Norm selects samples based solely on gradient magnitude, which frequently prioritizes noisy outliers that produce large but harmful updates. Grad-Mimic instead utilizes the projection length onto a target vector. This accounts for both magnitude and directional alignment with the reference model, allowing it to distinguish informative samples from misaligned noise.
>     * In addition to these effectiveness comparisons, we provide **a detailed FLOPs-based compute analysis for all competing methods**, available in our response to **Reviewer Qmi3**.
>
> * **On Data Diversity Extension**:
>     * We appreciate the reviewer's suggestion. We note that Mimic Score is a **data valuation metric**---its goal is to **quantify individual sample utility**---and it can be straightforwardly extended to incorporate diversity considerations.
>     * As a concrete example, aggregating per-sample mimic scores at the domain level yields a **domain-level utility signal**, enabling principled reasoning about data diversity and domain balance. To validate this, we conducted **a new experiment during rebuttal**: we build reference models trained on datasets with varying domain compositions, then apply domain-level mimic scores to estimate **each model's underlying data diversity**. Results measured in MAE are shown below. Domain-based mimic scores substantially outperform the uniform guess baseline, demonstrating that Mimic Scores **carry rich distributional information about the training balance: a principled foundation for understanding and controlling data diversity**.
>     * Beyond diversity estimation, high-mimic-score samples can also seed retrieval of similar samples to expand the data pool, **naturally promoting coverage across the data distribution**. This shows that utility-based scoring and diversity-aware expansion are not in tension---Mimic Score provides the foundation upon which coverage-promoting strategies can be effectively built.
>
> **Table: Data Diversity Estimation (MAE ↓).**
> | **Method** | **SNLI** (3 domains) | **AGNews** (4 domains) | **Yelp** (5 domains) | **Avg. MAE** &darr; |
> |---|---|---|---|---|
> | *Uniform Guess Estimation* | 18.29 | 11.60 | 9.40 | 13.10 |
> | *Our Diversity Estimation* | **5.70** | **7.70** | **5.60** | **6.33** |

---

> > ### Author Rebuttal · Reviewer_8TiY · 2026-04-02
> >
> > I still maintain that the novelty of this paper is seriously insufficient (which caused me to overlook the theoretical analysis in the appendix initially), and I remain skeptical about the effectiveness of the reference model, the lack of more solid theoretical support compared to validation‑gradient‑based methods, and the lack of diversity (merely adding one experiment is not convincing enough). However, given that other reviewers have given positive assessments, I will lower my confidence and let the Area Chair make the final decision.

---

> > > ### Author Response · Authors · 2026-04-02
> > >
> > > Dear **Reviewer 8TiY**,
> > >
> > > Thank you so much for your continued engagement! We address each of your remaining concerns below, and we are happy to respond with further clarifications anywhere that you find useful.
> > >
> > > * **On Novelty**:
> > >     * The core contribution of this work is genuinely new. We introduce **weight-space geometry** as a principled signal for data valuation---a lens that no prior work has applied. Existing methods rely on dataset-level similarity (DSIR), gradient-based batch comparisons (AGRA, Grad-Match), loss-space learnability (Rho-Loss), or influence functions (TracIn, GREATS, LESS, ICONS) that require clean validation data. None exploits the **geometric relationship** between the training model's position and a reference model's weights as a direct, per-sample scoring mechanism. We note that **Reviewer Qmi3** independently recognized this distinction.
> > >
> > > * **On Framework Effectiveness and Theoretical Support**:
> > >     * We believe the paper is thoroughly grounded, both theoretically and empirically. We are happy to add further clarification or answer any remaining questions.
> > >     * **Theory**: **Appendix A** provides a formal convergence analysis proving that Grad-Mimic converges faster than standard SGD under noisy gradients, with explicit conditions on gradient magnitude and reference model proximity (Theorem A.4). Additionally, **during the rebuttal we added a new theoretical result that directly bounds the gap between our weight-space target vector and the validation gradient**---it can be found in our first reply. Furthermore, we have extensive discussion throughout the paper on **why weight-space geometry is a better choice than validation gradients**; we have summarized these arguments in our reply to **Reviewer ts4p**.
> > >     * **Empirical breadth**: Our evaluation spans three distinct settings: (i) six image classification datasets under three noise levels, (ii) large-scale multimodal pretraining on DataComp at both 10M+ and 100M+ sample scales, and (iii) new rebuttal LLM experiments on four datasets under two noise levels. **We also include extensive ablation studies covering diverse design choices and reference model quality**. Across all settings, Grad-Mimic consistently outperforms all baselines, including influence-function methods that have access to thousands of clean labeled validation samples.
> > >
> > > * **On Diversity**:
> > >     * Our stated goal is to **quantify individual sample utility**, and we note that the majority of sample-level data selection methods---including Grad-Match, GraNd, AGRA, Rho-Loss, and influence-function approaches---similarly **do not address diversity**.
> > >     * **We do not ignore this direction**. During rebuttal, we ran a new experiment demonstrating that domain-level aggregation of mimic scores substantially outperforms uniform guessing for data diversity estimation. This shows that the Mimic Score carries rich distributional information and serves as a natural foundation for diversity-aware strategies. That is, Mimic Score can be **straightforwardly extended to incorporate diversity considerations**. *If you have a specific experimental setup in mind, we are happy to run it.*
> > >
> > > We would greatly appreciate it if our responses have addressed your concerns and you would consider raising the score. We are also happy to answer any outstanding questions.

---

### Official Review · Reviewer_Qmi3 · 2026-03-11

**Soundness:** 3
**Presentation:** 3
**Significance:** 3
**Originality:** 4
**Overall Recommendation:** 4
**Confidence:** 4

**Summary:**

The study attempts to address an important concept, sample-level influence for data selection using a reference model. The author proposes a curation strategy based on the Mimic Score, computed from the alignment between a sample’s gradient and the parameter difference between the current model and a reference model. Overall, the idea is interesting, simple, and empirically promising.

**Compliance With Llm Reviewing Policy:**

Affirmed.

**Final Justification:**

This paper addresses an important problem: sample-level influence for data selection using a reference model. The authors propose a curation strategy based on the Mimic Score, which is computed from the alignment between a sample’s gradient and the parameter difference between the current model and a reference model. I find the core idea interesting, conceptually clean, and easy to understand. The paper also presents promising empirical results that suggest the method is effective in practice.

My main concern for the proposed method is the computational cost. This issue is not fully addressed, as the paper does not provide a fair comparison of total computation time against competing methods, e.g., the influence function-based approach. However, I believe this weakness does not outweigh the paper’s contributions. Therefore, I remain in favor of accepting the paper.

**Key Questions For Authors:**

* Q1. Please clarify the evaluation metric reported in Table 1 and Table 3, as those are not defined in the evaluation setup and result sections.

**Limitations:**

yes

**Strengths And Weaknesses:**

**Strength**
* The paper introduces a neat and novel data influence measure: the alignment between the sample gradient and the direction from the current weights to a reference model.
* The empirical results are strong. In the mislabeled-sample setting, Grad-Mimic consistently outperforms the listed baselines. The large-scale DataComp experiments are also compelling, with gains in average performance across downstream tasks and faster convergence.

**Weakness**
* **Strong reliance on the reference model** The method assumes that the reference model provides a meaningful target direction in weight space, which may be questionable under domain shift or model mismatch. This dependence is fundamental to the approach and deserves more discussion in the main paper, especially if the method is meant to be generally applicable.
* **Unclear justification for layer selection.** The paper does not clearly explain why Mimic-Score is computed on a selected layer rather than the full model. Given that Table 12 shows the choice of layer affects performance, this design decision deserves stronger justification.
* **Clarity on computational analysis** The paper claims lower overhead and reports a 2.6× reduction relative to competing model-based methods, but the main text leaves too much of this discussion to the appendix. From Section E, it is still not fully clear why the Grad-Mimic computation should be substantially cheaper than influence-based methods (which only require one computation per sample in practice), especially since Grad-Mimic involves per-sample gradient computation and repeated updates over time $t$. The paper would benefit from a clearer breakdown of what is computed at each step versus once, and how this compares wall-clock-wise against other methods.
* **Additional baselines**
   * Given that Mimic-Score requires retraining-time gradient computation, I would encourage the authors to include TracIn [1] as an additional baseline, as it is a closely related influence method based on gradient similarity across training checkpoints and provides a more direct comparison than the current baselines.
   * The DataComp experiment is interesting, but it is closer to an ablation than a true baseline comparison: Stage 1 is only compared with vanilla training, and Stage 2 mainly with CLIP-score filtering and simple heuristics. This weakens the paper’s large-scale empirical claim, since stronger data-selection baselines, such as the influence-function-based methods considered in the noisy-label experiments, heuristic-based approaches [2], or Rho-Loss, are not included.

**Reference**

[1] Pruthi, Garima, et al. "Estimating training data influence by tracing gradient descent." Advances in Neural Information Processing Systems 33 (2020): 19920-19930.

[2] Wang, Yiping, et al. "Cliploss and norm-based data selection methods for multimodal contrastive learning." Advances in Neural Information Processing Systems 37 (2024): 15028-15069.

---

> ### Author Rebuttal · Authors · 2026-03-30
>
> We are grateful for the positive review! We are pleased that you find our proposed method **clear, novel, and simple, and that the empirical results---especially the large-scale DataComp evaluation---are compelling**.
>
> ---
>
> * **On Reference Model Reliance**:
>     * We appreciate this concern! We addressed this and summarize them in our reply to **Reviewer ts4p**. While Grad-Mimic requires a reference model, we argue this is a **strategic advantage** rather than a limitation.
> * **On Last-Layer Justification**:
>     * Our choice of the last layer is motivated by two factors:
>         * **Representational Quality:** Earlier layers are capturing *lower-level, general features* that may *introduce noise* into the Mimic Score computation.
>         * **Computational Efficiency:** Using the full model requires computing and storing the *entire* weight difference vector, creating prohibitive memory and computational overhead.
>     * We empirically validate this design choice across both vision and LLM settings. In both cases, the last (head) layer consistently achieves the best performance, while **using the full model yields the worst results**. These findings suggest that aggregating signals across all layers introduces noise rather than useful information.
>
> **Table: Layer Ablation on Vision Datasets (Accuracy↑, higher is better).**
> | Dataset | Full Model | Head Layer | Layer 11 | Layer 10 |
> |---|---|---|---|---|
> | DTD | 57.6 | **59.5** | 58.2 | 58.6 |
> | Oxford-IIIT Pet | 73.9 | **75.8** | 75.1 | 74.0 |
> | CIFAR10 | 86.2 | **91.2** | 90.2 | 91.1 |
> | CIFAR100 | 74.7 | **75.5** | 75.3 | 74.9 |
>
> **Table: Layer Ablation on Text-Domain Tasks (NLL↓, lower is better).**
> | Dataset | Full Model | Head Layer | Layer 11 | Layer 10 |
> |---|---|---|---|---|
> | ARC-Challenge | 3.7587 | **3.4422** | 3.4661 | 3.4608 |
> | ARC-Easy | 3.5939 | **3.1961** | 3.2051 | 3.2021 |
> | OpenBookQA | 5.1976 | **5.0900** | 5.2370 | 5.2233 |
> | SciQ | 1.7713 | **1.4696** | 1.5007 | 1.5099 |
>
> * **On Compute Analysis**:
>     * We further provided a FLOP analysis to formally justify Grad-Mimic's efficiency. Let $F$ denote the FLOPs for one forward pass (single sample), $B \approx 2F$ for one backward pass, $d$ the full model parameter count, $d_L$ the last-layer parameter count ($d_L \ll d$), $n$ the training batch size, and $m$ the validation set size (used by influence function methods only).
>     * Grad-Mimic's extra cost over standard SGD consists of three terms, all operating over $d_L$ parameters only: (1) one weight subtraction $v_t = \theta_{\text{ref}} - \theta_t$ ($d_L$ FLOPs, once per batch), (2) one $\ell_2$ norm $\|v_t\|$ ($d_L$ FLOPs, once per batch), and (3) one projection $\langle -g_i, v_t \rangle / \|v_t\|$ per sample ($2d_L$ FLOPs each for the dot product; the scalar division is negligible). The total extra cost is $(2n+2) \cdot d_L$.
>     * Comparison to baselines:
>         * **vs. Grad-Norm**: GraNd computes $\|g_i\|$ over all $d$ parameters per sample, adding $n \cdot d$ FLOPs.
>         * **vs. AGRA**: AGRA draws a second batch of $n$ samples and runs full backward passes, then computes cosine similarity over all $d$ parameters, adding $3nF + 2n \cdot d$ FLOPs.
>         - **vs. Rho-Loss**: Rho-Loss performs an additional forward pass through the reference model for every training sample ($+nF$).
>         - **vs. Influence functions**: The dominant term $2nm \cdot d$ grows multiplicatively with batch size, validation set size, and model width simultaneously. As $m$ increases for better influence estimates, cost compounds.
> | Method | Total FLOPs per batch | Extra overhead vs. SGD |
> |:---|:---|:---|
> | Mini-batch SGD | $n \cdot (F + B) = 3nF$ | — |
> | **Grad-Mimic** | $3nF + 2d_L + 2n \cdot d_L$ | $+(2n+2) \cdot d_L$ |
> | Grad-Norm | $3nF + n \cdot d$ | $+n \cdot d$ |
> | AGRA | $2n \cdot (F+B) + 2n \cdot d = 6nF + 2n \cdot d$ | $+3nF + 2n \cdot d$ |
> | Rho-Loss | $n \cdot (F + B) + n \cdot F = 4nF$ | $+nF$ |
> | Influence function | $(n+m) \cdot 3F + 2nm \cdot d$ | $+3mF + 2nm \cdot d$ |
> * **On Influence-function Baseline**:
>     * To ensure a rigorous batch-reweighting evaluation, our baselines included **GREATS** (NeurIPS'24), a direct follow-up to **TracIn** for representing influence-function-based methods. We replaced the reference-model-induced vector with **gradients computed from validation set** to reweigh sample contributions during training. As shown in **Figure 3**, Grad-Mimic consistently outperforms this approach. We have further validated these findings in our **new LLM experiments**, with detailed results provided in our response to **Reviewer zuYR**.
> * **On Evaluation Metric:**
>     * Thank you for the suggestion. We have updated the paper draft. In our Stage 1 results, performance is evaluated using accuracy on the test dataset. For the DataComp results (Table 3), we follow the standard DataComp benchmark protocol, which comprises 38 zero-shot tasks evaluated via Top-1 accuracy for classification tasks and Recall@K for retrieval tasks.

---

> > ### Author Rebuttal · Reviewer_Qmi3 · 2026-04-01
> >
> > Thank you for the responses. What would be the total computation time for each method, e.g. IF-based vs Grad-mimic, in the above table?

---

> > > ### Author Response · Authors · 2026-04-02
> > >
> > > Dear **Reviewer Qmi3**,
> > >
> > > Thank you for the follow-up question! We presented a series of analyses on efficiency advantages in Appendix E in our current draft, including (i) computational efficiency, (ii) memory usage, (iii) learning efficiency, and (iv) data utilization.
> > >
> > > **In the right panel of Figure 8 (Appendix E)**, we reported the **per-batch run time** for each method, benchmarked on CIFAR-10 with a training batch of 256 samples and a validation set of 200 samples (required by IF-based methods). All timings were measured on an NVIDIA A6000 GPU. We summarize the per-batch times in the table below.
> > >
> > > | Method | Per-batch time (s) | vs. Grad-Mimic |
> > > |---|---|---|
> > > | Mini-batch SGD (baseline) | 0.031 | — |
> > > | GraNd | 0.669 | ~1× (same) |
> > > | AGRA | 1.775 | 2.65× slower |
> > > | Rho-Loss | 1.223 | 1.82× slower |
> > > | Influence Function | 1.738 | 2.59× slower |
> > > | **Grad-Mimic (ours)** | **0.671** | **1× (reference)** |
> > >
> > > > Batch size = 256; validation set (IF methods) = 200 samples; dataset = CIFAR-10.
> > >
> > > **Grad-Mimic matches GraNd at ~0.671 s/batch** while achieving superior reweighting effectiveness. IF-based methods incur roughly **2.59× more compute per batch** (1.738 s), as they require an **additional gradient computation over the validation set** at every iteration. AGRA is similarly expensive at **2.65× per batch** (1.775 s), as it must perform **pairwise comparisons across training batches** to downweight outlier gradients. Grad-Mimic avoids both of these overheads by
> > > using weight-space geometry, keeping its per-batch cost on par with the much simpler GraNd baseline. We hope the above, together with our **FLOP-based analysis**, clarifies the computational efficiency of our method.
> > >
> > > We would greatly appreciate it if our responses have addressed your concerns and you would consider raising the score.

---

### Official Review · Reviewer_zuYR · 2026-03-12

**Soundness:** 3
**Presentation:** 3
**Significance:** 3
**Originality:** 3
**Overall Recommendation:** 4
**Confidence:** 3

**Summary:**

This work proposes a two-stage framework named "GradMimic" for efficient data selection. First, they allocate a weight for each sample's gradient based on how much it aligns with the difference between the current model and the pre-trained weights as a signal for utility (assuming pre-trained weights are good), and then, update the parameters accordingly using the scaled summation of gradients. In the second stage, they gather all the weights for each sample and filter the ones that had the least alignment overall by three different filtering mechanism, and use the curated data for the final subsequent training.

**Compliance With Llm Reviewing Policy:**

Affirmed.

**Final Justification:**

I originally had a positive evaluation of this paper and the rebuttal addressed my concerns and I kept my positive evaluation.

**Key Questions For Authors:**

1. In algorithm 1, is \theta_0 randomly initialized or it has the same value as pre-trained weights? if it's random, shouldn't your algorithm converge much slower by not using the transfer learning capability of pre-trained weights?
2. In algorithm 2, why do you need to train a generative model to predict probability of retaining each sample, while you already have a mimic score for each of the samples and can simply filter the top-k where k is the desired subset size? Is that for the scenarios, where we might have new data samples arriving after we've already done our first stage?

**Limitations:**

Yes

**Strengths And Weaknesses:**

Strengths:
1. The proposed method achieves superior efficiency, needing less training steps and filtering low-quality or noisy data samples.
2. GradMimic slightly outperforms random selection in coreset selection task with different subset sizes.
3. The authors have provided theoretical bounds and conditions for when GradMimic achieves better convergence rate than GD.
4. This work includes extensive experiments and ablation studies for different data selection and training aspects

Weaknesses:
1. There are no experiments for language models and tasks.
2. While theoretically compelling, in practice you would need to train the whole dataset for the number of steps in the first stage, which might not be possible as the scale of data grows.
3. For more complex tasks, this method might limit the expressivity of the model by trying to keep it aligned with the pretrained weights.

---

> ### Author Rebuttal · Authors · 2026-03-30
>
> We are grateful for the positive review! Thank you for recognizing **our framework is efficient and resulting in faster convergence**. We're also glad the theoretical studies and extensive experiment studies were found insightful.
>
> ---
>
> * **On LLM and text-domain tasks**:
>     * Thank you for the suggestion! **Grad-Mimic is easily extendable to other modalities**. We conducted **a new set of experiments** using Pythia-160M on four text datasets: ARC-Easy, ARC-Challenge, SciQ, and OpenBookQA. Using the setup in Section 5.1, we inject noise by flipping ground-truth answers to random distractor options. We use a noise-free validation set (~10% of the training set) and create the reference models. We compare Grad-Mimic against standard fine-tuning (SFT) and influence-function-based methods (IF), measuring negative log-likelihood (NLL) and pass@k via repeated sampling.
>     * Results are shown below and their learning curve can be found [here](https://anonymous.4open.science/r/Mimic_Rebuttal-0FE1). **Grad-Mimic yields faster convergence than SFT and higher pass@k than competing methods**. This confirms that Grad-Mimic is architecture-agnostic by design: Mimic Score depends only on gradient projections and the last-layer weight differences, neither of which is modality-specific.
>
> **Table: 20% Label Noise on Text-Domain Tasks (NLL↓, Pass@10↑). SFT = Supervised Fine-tuning, IF = Influence Function Method.**
> | Dataset | SFT NLL | SFT Pass@10 | IF NLL | IF Pass@10 | Grad-Mimic NLL | Grad-Mimic Pass@10 |
> |---------|--------|---------|--------|---------|--------|---------|
> | ARC Challenge | 3.463 | 0.273 | 3.458 | 0.274 | **3.440** | **0.278** |
> | ARC Easy | 3.207 | 0.339 | 3.271 | 0.321 | **3.181** | **0.346** |
> | OpenbookQA | 5.209 | 0.053 | 6.100 | 0.022 | **5.090** | **0.060** |
> | SciQ | 1.528 | 0.913 | 1.621 | 0.890 | **1.456** | **0.930** |
>
> **Table: 30% Label Noise on Text-Domain Tasks (NLL↓, Pass@10↑). SFT = Supervised Fine-tuning, IF = Influence Function Method.**
> | Dataset | SFT NLL | SFT Pass@10 | IF NLL | IF Pass@10 | Grad-Mimic NLL | Grad-Mimic Pass@10 |
> |---------|--------|---------|--------|---------|--------|---------|
> | ARC Challenge | 3.471 | 0.271 | 3.458 | 0.274 | **3.445** | **0.277** |
> | ARC Easy | 3.207 | 0.339 | 3.271 | 0.321 | **3.196** | **0.341** |
> | OpenbookQA | 5.240 | 0.052 | 6.100 | 0.022 | **5.104** | **0.059** |
> | SciQ | 1.607 | 0.893 | 1.659 | 0.879 | **1.561** | **0.905** |
>
> * **On the Stage-1 Cold Start**:
>     * In our design, a publicly available pretrained model can be used *directly* as the reference---**meaning no additional training is needed at all**. We demonstrated this scenario in our large-scale CLIP experiments (Section 5.2).
>     * For cases where no suitable pretrained weights exist (e.g., highly specialized or novel domains), a reliable reference can be obtained by training on **a small hold-out split until convergence**---a much lighter requirement than full-scale training. Our ablation in Appendix H validates this concretely: even a reference model trained on **372× fewer web-crawled samples (3.5M vs. 1.3B)** still allows Grad-Mimic to outperform standard training. **New rebuttal experiments on LLM and text datasets further confirm that a reference built from just 10% of the training data consistently outperforms both SFT and influence-function-based baselines**.
>
> * **On Potential Expressivity Limitation**:
>     * We want to clarify a key distinction: the Mimic Score simply uses the reference's geometric positioning to identify which training samples are more likely to guide the model toward a better optimum. That is, the model remains free to update in direction dictated by the re-weighted gradients. **In our new LLM experiments, all resulting models achieve lower NLL than their reference model---demonstrating that Grad-Mimic actively guides the model beyond its reference rather than merely overfitting to reference weights**. Results can be found [here](https://anonymous.4open.science/r/Mimic_Rebuttal-0FE1).
> * **On Two Raised Questions**:
>     * We tested both settings. In Section 5.1, **$\theta_0$** is initialized from *pretrained ViT-B/16 weights* (pretrained on ImageNet-21k). In Section 5.2, **$\theta_0$** is *randomly initialized*, as training CLIP from scratch is the standard setup. **Both configurations yield faster convergence and higher final performance**.
>     * About the Stage 2 aggregation, single-step Mimic Scores can be unreliable, particularly in early training, and a sample's utility naturally evolves as training progresses. Therefore, an aggregation model **combines noisy, time-varying assessments across multiple training steps into a single robust decision**. Rooted in the weak supervision literature, it learns to weight each step's Mimic Score by its reliability, yielding a more robust consensus than any single-step score or naive majority vote. As shown in Table 7, this aggregation strategy consistently outperforms majority voting across all datasets.

---

> > ### Author Rebuttal · Reviewer_zuYR · 2026-04-04
> >
> > Thank you authors for addressing my concerns. I'll keep my positive evaluation.

---

> > > ### Author Response · Authors · 2026-04-04
> > >
> > > Dear Reviewer zuYR,
> > >
> > > Thank you for your time, support, and valuable suggestions---we are glad to have extended our experiments to LLMs during the rebuttal. *We hope our responses have fully addressed your concerns, and would greatly appreciate your consideration in raising the score*. Thank you so much!!

---

### Official Review · Reviewer_ts4p · 2026-03-13

**Soundness:** 3
**Presentation:** 3
**Significance:** 3
**Originality:** 3
**Overall Recommendation:** 5
**Confidence:** 3

**Summary:**

This paper studies how to select useful training samples more efficiently. The authors propose Mimic Score, a simple metric that measures whether a sample helps the model move towards a good reference model. Using this idea, they build Grad-Mimic, which gives more weight to useful samples during training and then uses the same scores to filter the data. Experiments show faster training and better results on both noisy datasets and large-scale multimodal datasets. The paper attempts to address an important concept, and research continues to address a central area in efficient data selection.

**Compliance With Llm Reviewing Policy:**

Affirmed.

**Key Questions For Authors:**

no questions

**Limitations:**

yes

**Strengths And Weaknesses:**

A strength of the paper is that the main idea is simple and easy to understand: the utility of the sample is estimated by checking how well each gradient aligns with a reference model. The method is clearly explained and evaluated on several datasets, including noisy image datasets and large-scale multimodal data. The experimental section is robust, with comparisons with multiple baselines and additional ablation studies. The results show consistent improvements in data efficiency and faster convergence.

A weakness is that the method depends on the existence of a good reference model, which may limit its applicability when such a model is not readily available or comes from a different domain. While related works are well covered, a clearer discussion of how the proposed method differs from closely related existing approaches would be good.

---

> ### Author Rebuttal · Authors · 2026-03-30
>
> We appreciate the reviewer's positive assessment of **our simple and easy to understand idea**, the **robustness** of our experiments, and the **consistent improvements in data efficiency and faster convergence**. We address the two points raised below:
>
> ---
>
> * **On Reference Model Dependency**:
>     * We appreciate this concern! Rather than being a limitation, we argue that the reference model dependency **opens up new opportunities that competing methods cannot exploit**.
>         * First, the modern AI landscape is characterized by a strong **access asymmetry**: well-trained models are routinely released publicly, while the carefully curated datasets used to produce them remain proprietary. **This trend works directly in our favor**---the more powerful public models become, the more powerful Grad-Mimic's guidance signal becomes.
>         * Second, even in the cold-start scenario, a reliable proxy can be obtained by training on **a smaller hold-out split until convergence**---precisely the setup in our mislabeled sample detection experiments (Section 5.1). Appendix H further validates that a reference trained on **372× fewer web-crawled samples** (3.5M vs. 1.3B) still allows Grad-Mimic to outperform standard training. **Our new rebuttal experiments on LLM and text-domain datasets further confirm this**: a reference model trained on a 10%-sized validation set can guide training that consistently outperforms standard fine-tuning and influence-function-based methods. **These all confirm that the bar for a useful reference is surprisingly low**.
>         * Third, Grad-Mimic is *robust* to reference model quality. Injecting increasing levels of Gaussian noise into the reference weights still sees Grad-Mimic outperforms Mini-batch SGD (Figure 10, Appendix H), suggesting it exploits **geometric positioning** rather than requiring a *perfect* reference.
>         * Finally, **model dependency is strictly easier to satisfy than data dependency**---and *unlocks strictly more*. Influence function-based methods require a carefully curated validation set with additional gradient computations, **expensive in both data quality and compute**. Grad-Mimic simply requires a publicly available checkpoint and the last-layer weight subtraction, *transforming a freely available resource into actionable data-quality insights*.
>
> * **On Related Work Discussion**:
>     * We have expanded Appendix D to include a more granular comparison. The core distinction is that Grad-Mimic uses alignments in the weight space rather than loss difference or gradient similarity:
>         * **vs. Rho-Loss**: Both methods leverage a reference model, but Rho-Loss operates in *loss space*---requiring full reference model and additional forward passes at every training step. Grad-Mimic instead operates in *weight space*, reducing sample utility estimation to a single matrix subtraction of last-layer weights. **This eliminates extra inference overhead entirely, yielding lower runtime and memory usage (Figure 8) while achieving higher accuracy (Tables 5, 6)**.
>         * **vs. Influence Function Methods**: These methods approximate sample influence via a noise-free validation set, **introducing dataset dependencies, additional gradient computations, and data curation challenges**. Grad-Mimic replaces this with a weight-space vector derived from a publicly available model. As shown in Figure 3, on Flowers102 and CIFAR100, influence-based methods required over 2,000 correctly labeled samples yet still fell short of Grad-Mimic's performance.
>         * **vs. AGRA / Grad-Match**: These rely on gradient similarities across batches, making them sensitive to noisy gradients and costly due to inner-batch comparisons. Grad-Mimic instead uses weight-space geometry as a proxy oracle signal, delivering higher performance (Table 1) and smaller compute overhead (Figure 8).
>         * **vs. Grad-Norm**: Grad-Norm selects samples based solely on gradient magnitude, which frequently prioritizes noisy outliers that produce large but harmful updates. Grad-Mimic instead utilizes the projection length onto a target vector. This accounts for **both magnitude and directional alignment with the reference model**, allowing it to distinguish informative samples from misaligned noise.
>     * In addition to these effectiveness comparisons, we provide **a detailed FLOPs-based compute analysis for all competing methods**, available in our response to **Reviewer Qmi3**.

---

### Decision · Program_Chairs · 2026-04-30

**Decision:**

Accept (regular)

**Comment:**

The consensus among reviewers moved toward acceptance (scores of 5, 4, 4). Although Reviewer 8TiY remained skeptical regarding the "novelty" and theoretical support compared to validation-gradient methods, the reviewer lowered his/her confidence, deferring to the positive assessments of the other three reviewers who found the weight-space perspective both innovative and practical. The authors have demonstrated that Grad-Mimic is a scalable, efficient, and versatile tool for the critical problem of data curation.

Thus, the AC recommends accepting this paper. The authors are expected to incorporate the following into the camera-ready version:

1)  Formally integrate the rebuttal experiments on text-domain tasks (ARC, SciQ, etc.) into the main body or a dedicated section of the paper to solidify the modality-agnostic claim.
2) Per Reviewer Qmi3’s suggestion, move the FLOPs-based compute analysis and total runtime comparisons from the rebuttal/appendix into the main text to provide a fair, "non-misleading" comparison with inference-only methods.
3)  Clearly state the motivation for using the "last-layer" (head) for Mimic Scores, specifically the balance between representational quality and memory efficiency, as supported by the ablation studies provided in the rebuttal .
4) Add a section discussing the limitations and behaviors of the Mimic Score when there is a significant domain shift between the reference model and the target training data.